# High Accuracy Power Quality Evaluation under a Colored Noisy Condition by Filter Bank ESPRIT

**Elaine Santos [1], Mahdi Khosravy [1,2,\*], Marcelo A. A. Lima[1], Augusto S. Cerqueira [1], Carlos A. Duque [1] and Atsushi Yona [3]**

[1] Electrical Engineering Department, Federal University of Juiz de Fora, Juiz de Fora, MG 36036-900, Brazil; elaine.santos@engenharia.ufjf.br (E.S.); marcelo.lima@engenharia.ufjf.br (M.A.A.L.); augusto.s.cerqueira@gmail.com (A.S.C.); carlos.duque@ufjf.edu.br (C.A.D.)

[2] Media Integrated Communication Lab, Graduate School of Engineering, Osaka University, Osaka 565-0871, Japan

[3] Electrical Engineering Department, University of the Ryukyus, Okinawa 903-0213, Japan; yona@tec.u-ryukyu.ac.jp

\* Correspondence: dr.mahdi.khosravy@ieee.org

**Abstract:** Due to the highly increasing integration of renewable energy sources with the power grid and their fluctuations, besides the recent growth of new power electronics equipment, the noise in power systems has become colored. The colored noise affects the methodologies for power quality parameters' estimation, such as harmonic and interharmonic components. Estimation of signal parameters via rotational invariance techniques (ESPRIT) as a parametric technique with high resolution has proven its efficiency in the estimation of power signal components' frequencies, amplitudes, and phases for quality analysis, under the assumption of white Gaussian noise. Since ESPRIT suffers from high computational effort, filter bank ESPRIT (FB-ESPRIT) was suggested for mitigation of the complexity. This manuscript suggests FB-ESPRIT as well for accurate and robust estimation of power signal components' parameters in the presence of the colored noise. Even though the parametric techniques depend on the Gaussianity of contaminating noise to perform properly, FB-ESPRIT performs well in colored noise. The FB-ESPRIT superiority compared with the conventional ESPRIT and MUSIC techniques was demonstrated through many simulations runs on synthetic power signals with multiple harmonics, interharmonics, and subharmonic components in the presence of noises of different colors and different SNR levels. FB-ESPRIT had a significant efficiency superiority in power quality analysis with a wide gap distance from the other estimators, especially under the high level of colored noise.

**Keywords:** power quality; spectrum estimation; ESPRIT; FB-ESPRIT; MUSIC; filter bank; colored noise

## 1. Introduction

Electrical power quality (EPQ) analysis [1,2] considerably assists power system planners and designers in hybrid energy storage loss mitigation [3], renewable energy monitoring [4,5], observing the disturbances in power quality [6], proper connection of electric vehicles to the grid [7], etc. EPQ gains higher importance, especially in this era of renewable and green energy, where the power industries extend the use of renewable power sources like wind turbines, solar power plants, hydroelectricity, etc., and each brings its effect of nature-inherited uncertainty to the system. To have a secured and qualified integration of renewable sources to the grid, EPQ should be continuously monitored and analyzed in terms of disturbances and the introduction of harmonics, sub-harmonics, and interharmonics to the power content [1]. As the main characteristic of renewable energy sources is their dependency on

nature, their oscillating production introduces a special kind of noise to the power system. For example, in the case of a wind turbine, it introduces a low frequency noise with a frequency band of $10^{-1}$ to $5 \times 10^{-1}$ Hz due to wind fluctuations [8]. As a matter of fact, this type of noise has a colored nature, as it covers a portion of the power bandwidth, but not all of it. Besides the increasing growth of renewables and their introduced uncertainties to the system, power electronics' proliferation in new technologies also bring their type of colored noise to the system.

In the context of the analysis of electrical power quality, the estimation of harmonics, interharmonics, and sub-harmonics is crucial, as misadjustment of these parameters can cause severe issues in the grid and large losses [9]. The main harmonic estimation methods are separated into two major groups: nonparametric methods and parametric methods. The algorithms initially used in the field of power systems were nonparametric methods, for example the fast Fourier transform (FFT). The IEC 61000-4-7 standard recommended FFT as it has a higher computational efficiency than other methods, but it ends up having a problem known as spectral leakage [10,11]. The first parametric method of subspace auto-decomposition was that of Pisarenko [12]. In terms of spectrum estimation with higher resolution, it surpasses FFT. Pisarenko method's disadvantage lies in the need for the exact pre-knowledge of the model order. Furthermore, it results in inaccuracy due to the statistical estimation of the autocorrelation lag. Prony has been the method preferred over Pisarenko as it possesses lower complexity [13], but still is not efficient in complexity terms and susceptible to noise. The subspace division technique was firstly used in the MUSIC [14,15] algorithm, which has the capability of performing over a short duration of the signal and giving high resolution estimation [16]. However, it requires a big size of operating memory and is computationally heavy [17].

Later on, the spectrum estimation using the rotational invariance characteristic of the signal (ESPRIT) was introduced [17–21]. ESPRIT is an efficient spectrum estimation technique with high precision without the need for memory storage [22]. Amongst the parametric spectrum estimation techniques [23,24], because of the above mentioned advantages of ESPRIT, especially high resolution, it has been widely used as partially listed below. The literature reports ESPRIT's efficiency in islanding detection using the generator frequency [25], identification of low frequency modes in a power system [26], power quality indexing in the IEEE standard of 1459–2010 [27], power quality measurement [28], fast power system harmonic analysis under non-stationary conditions [29], dynamic phasor estimation [30], harmonic parameters' estimation under multiplicative noise [31], assessment of distortions in powerline waveforms [32], synchronized phasor monitoring [33,34], distribution system harmonic source identification [35], etc.

Besides all the advantages, as a subspace-based parametric method, ESPRIT heavily relies on the assumption of the whiteness of any contaminating noise, and its estimation efficiency is degraded when the signal components are contaminated with colored noise [36]. Furthermore, it demands a high computational effort. The work in [37] overcame its complexity by the association of ESPRIT with a filter bank (FB-ESPRIT). This manuscript demonstrates that the association of the filter bank not only mitigates the complexity of ESPRIT, but also results in higher efficiency in the parameters' estimation of a signal contaminated by different types of colored noise (pink, red, blue, and violet).

The organization of the paper is as follows. Section 2 explains the colored noise issue. Section 3, after a brief explanation of ESPRIT, provides the theoretical justification of FB-ESPRIT's performance under colored noise. Section 4 analyzes the efficiency of FB-ESPRIT in terms of accuracy and a robustness comparison to conventional ESPRIT and MUSIC under colored noise. Finally, Section 5 presents the concluding remarks.

## 2. Colored Noise and Parametric Estimation

The problem of parameter estimation in a signal with colored noise has been reported in the literature, with some of the resolving efforts listed below. The work in [38] performed prewhitening techniques so that they could recover harmonics, and in [39], the colored noise was suppressed by the extension of the characteristic equation of the harmonic recovery. The authors described a method for

eliminating false peaks of spectral estimation and compared their method with the MUSIC method and the cross-spectral Levinson approach. The work in [40] deployed numerical differentiation to track the frequency of the power signals corrupted in colored noise, and The work in [41] used a high-resolution MUSIC 2D algorithm to retrieve two-dimensional harmonic frequencies contaminated with colored noise. Finally, The work in [42] investigated the resonance induced by colored noise at the subharmonic frequencies and concluded that the type of colored noise has important effects on the system response.

To conceptualize the white noise, its power was considered distributed uniformly in the frequency spectrum ($\varphi_w(f) = N_w$, i.e., cte). According to some authors [43–45], white noise is the result of an analogy to the electromagnetic spectrum in the light range (where white light contains all the frequencies of the visible spectrum).

Table 1 shows the classification of noise by the power spectral density (PSD) of the noise. Please note that it lists some of the most well-known noise types and classifies them according to their PSD equivalent to the energy of the noise. The only one with the constant PSD is white Gaussian noise as it is the most important in the signal processing area [46]. In the case of this work, specifically, the analysis colored noises are pink, red, blue, and violet noises. Note that violet and blue noises have a power spectral density proportional to the signal frequency; however, the energy ratio of blue noise is more intense than violet noise. Pink and red noises are inversely proportional to the signal frequency. They have equivalent energy at the lower frequencies. At higher frequencies, red noise has less energy than pink noise. Figure 1 shows the PSD of the colored noises used in this work. White and pink noises are considered to be the most important noises in nature since they have the property of being noise with a Gaussian distribution (null mean value). The other noises (with other distributions) are produced artificially [46].

**Table 1.** Classification of noise according to the power spectral density.

| Relationship to PSD | Generic Name | Example of Noise |
|---|---|---|
| Constant | White Noise | Thermal |
| Proportional to $\frac{1}{f}$ | Pink Noise | Flicker |
| Proportional to $\frac{1}{f^2}$ | Brown or Red Noise | Popcorn |
| Proportional to $f$ | Blue Noise | x |
| Proportional to $f^2$ | Violet Noise | x |
| Proportional to $\frac{1}{f^{2.7}}$ | No Generic Name | Galactic Noise |
| Irregular form | No Generic Name | Atmospheric Noise |

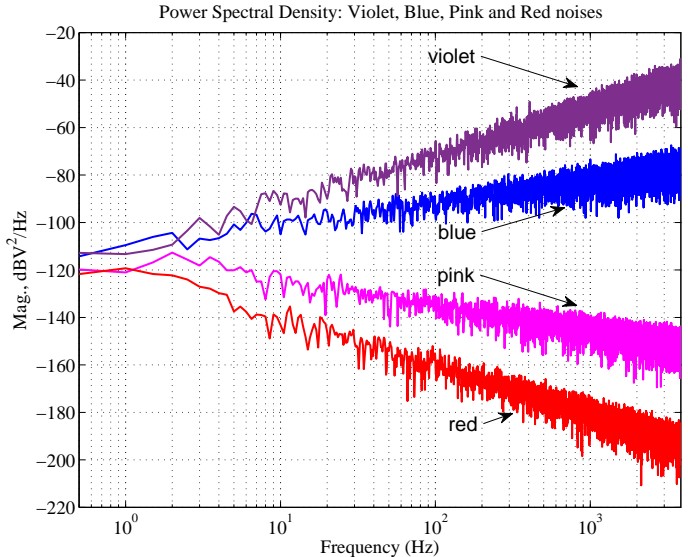

**Figure 1.** Power spectral density of violet, blue, pink, and red noises.

As already mentioned, parametric methods depend on the hypothesis that the noise is white [36]. These parametric methods (based on the spectral model) yield results with super-resolution, but cannot reconstruct the continuous or distributed part of the power spectral density (PSD), which is related to color noise [36]. The distributed power of the white noise is uniform in its frequency spectrum, and its energy potential is much smaller compared to the energy potential of the main components of the signal. Therefore, when using the ESPRIT technique, the noise subspace is discarded (denoising), and having a white noise, there will be no problem, since it is certain that the noise will actually be discarded. Now, having noise with a non-uniform power distribution in the frequency spectrum, there is no guarantee when it is necessary to discard the noise; all of it will be discarded.

## 3. Filter Bank ESPRIT

Filter bank ESPRIT (FB-ESPRIT) is a methodology to apply ESPRITs individually to the spread spectrum of the signal sub-bands [37]. It initially applies a filter bank to the signal and transfers it to the equally-spaced sub-bands, then each sub-band signal spectrum is spread by downsampling, and the conventional ESPRIT is applied to each spread spectrum sub-band individually. In this parallel process, each of the sub-bands has their estimated parameters separately by ESPRIT, and the signal estimated parameters are indeed the union of all sub-band ESPRITs applied together. Here, we give a brief explanation of ESPRIT and its association with the filter bank.

### 3.1. ESPRIT

The estimation of signal parameters via rotational invariance technique (ESPRIT) is a parametric spectrum estimation methodology introduced by Roy et al. [17–19]. Rotational invariance is a characteristic used for exploration between subspaces in ESPRIT, and therefore estimating the signal parameters.

Considering the Equation (1) signal model of the summation of complex exponentials and keeping in mind for each single tone complex exponential component of the signal $s_0[n] = e^{j2\pi f n}$, the time-shifting theory is applicable as follows:

$$s_0[n+1] = s_0[n]e^{j2\pi f}. \tag{1}$$

Then, the signal vector can be modeled as follows

$$\boldsymbol{x}[n] = \sum_{p=1}^{P} \alpha_p \boldsymbol{v}(f_p)e^{j2\pi n f_p} + \boldsymbol{w}[n] = \boldsymbol{V}\boldsymbol{\Phi}^n\boldsymbol{\alpha} + \boldsymbol{w}[n] \tag{2}$$

where $\boldsymbol{x}$ is a signal vector of length $M$ contaminated by the same size of noise vector $\boldsymbol{w}[n]$. $\boldsymbol{V}$ is the $(M \times P)$ matrix of time-frequency vectors $\boldsymbol{v}(f_p) = [1\ e^{j2\pi f_p}\ \dots\ e^{j2\pi(M-1)f_p}]^T$ for each of the $P$ frequencies as $\boldsymbol{V} = [\boldsymbol{v}(f_1)\ \boldsymbol{v}(f_2)\ \dots\ \boldsymbol{v}(f_p)]$. $\boldsymbol{\alpha} = [\alpha_1\ \alpha_2\ \dots\ \alpha_P]^T$ is the vector of exponential magnitudes $\alpha_p$s. The matrix $\boldsymbol{\Phi}$ is diagonal with phase shifts between the adjacent samples of the signal as:

$$\boldsymbol{\Phi} \quad = \quad diag\{\phi_1\ \phi_2\ \dots \phi_P\}$$

where $\phi_i = e^{j2\pi f_i}$. The ESPRIT approach to estimate the components' parameters is by obtaining the rotation matrix $\boldsymbol{\Phi}$.

Considering two $(M-1)$-length vectors $\boldsymbol{s}_{M-1}[n]$ and $\boldsymbol{s}_{M-1}[n+1]$ inside the $M$-length vector $\boldsymbol{s}_M[n]$ as follows:

$$\boldsymbol{s}_{M-1}[n] = \begin{bmatrix} s[n] \\ s[n+1] \\ \vdots \\ s[n+M-2] \end{bmatrix} \qquad \boldsymbol{s}_{M-1}[n+1] = \begin{bmatrix} s[n+1] \\ s[n+2] \\ \vdots \\ s[n+M-1] \end{bmatrix} \tag{3}$$

$s_{M-1}$ is a length $(M-1)$ and can be expressed as:

$$s_{M-1}[n] = V_{M-1}\Phi^n\alpha \tag{4}$$

$V_{M-1}$ is the same as $V$, with the difference that it is made of time-window vectors of an $M-1$ length as $V_{M-1} = [v_{M-1}(f_1)\ v_{M-1}(f_2)\ \ldots\ v_{M-1}(f_P)]$.

From Equation (4), we have the following definitions:

$$V_1 = V_{M-1}\Phi^n \qquad\qquad V_2 = V_{M-1}\Phi^{n+1} \tag{5}$$

$$V_2 = V_1\Phi \tag{6}$$

$$V\Phi = \begin{bmatrix} V_1 \\ \cdots \end{bmatrix} = \begin{bmatrix} \cdots \\ V_2 \end{bmatrix}. \tag{7}$$

On the other hand, we have the following relation between $V$ and the autocorrelation matrix of $x[n]$:

$$R_x = E\{x[n]x^H[n]\} = R_s + R_w \tag{8}$$

$$= \sum_{p=1}^{P} |\alpha_p|^2 v(f_p)v^H(f_p) + \sigma_w^2 I = VAV^H + \sigma_w^2 I \tag{9}$$

where $R_x$ is the autocorrelation matrix of signal vector $x[n]$, $R_s = VAV^H$ is the autocorrelation matrix of $s[n]$, and $R_w = \sigma_w^2 I$ is the autocorrelation of the white noise vector $w[n]$. The diagonal matrix $A$ is diag$\{|\alpha_1|^2\ |\alpha_2|^2\ \ldots\ |\alpha_P|^2\}$ and contains the powers of the corresponding complex exponentials. In practice, $R_x$ estimation is done by using the matrix $X$ as follows:

$$\hat{R}_x = \frac{1}{N}X^H X \tag{10}$$

where $X$ is the data matrix made as:

$$X_{N\times M} = \begin{bmatrix} x[0] & x[1] & \ldots & x[M-1] \\ x[1] & x[2] & \ldots & x[M] \\ \vdots & \vdots & \ddots & \vdots \\ x[N-1] & x[N] & \ldots & x[N+M-2] \end{bmatrix} \tag{11}$$

As the the first step in the ESPRIT process, it constructs the data matrix, then takes the singular value decomposition as follows:

$$X = L\Sigma U^H \tag{12}$$

where $L$ and $U$ are unitary matrices. $U$ forms an orthonormal basis for the $M$-dimensional signal vector space. At this stage, ESPRIT divides the basis into two subspaces for the signal and noise as:

$$U = [U_s|U_w] \tag{13}$$

where $U_s$ contains the $P$ biggest singular values corresponding to the $f = f_1, f_2, \ldots, f_P$ signal frequency components. Since $V$ and $U_s$ have a common subspace basis, there is an invertible transformation $T$ mapping from $U_s$ into $V$:

$$V = U_s T. \tag{14}$$

Similar to Equation (7) in partitioning $V\Phi$ into $(M-1)$-dimensional subspaces, ESPRIT partitions $U_s$ as:

$$U_s = \begin{bmatrix} U_1 \\ \cdots \end{bmatrix} = \begin{bmatrix} \cdots \\ U_2 \end{bmatrix} \tag{15}$$

As in Equation (14), there is the following relation between the two subspaces.

$$V_1 = U_1T \qquad V_2 = U_2T \tag{16}$$

As in Equation (6) between $V_1$ and $V_2$, there is the following relation between $U_1$ and $U_2$, but by a different rotation matrix:

$$U_2 = U_1\Psi. \tag{17}$$

SVD on data matrix $X$ gives the subspaces $U_1$ and $U_2$. Then, using a least squares technique, we obtain $\Psi$ from Equation (17):

$$\Psi = (U_1^H U_1)^{-1} U_1^H U_2 \tag{18}$$

From Equations (6), (16), and (17), it can be obtained that:

$$\Psi T = T\Phi. \tag{19}$$

This indicates that the elements on the diagonal of $\Phi$ are the eigenvalues of $\Psi$; thus, ESPRIT estimates the frequencies as:

$$\hat{f}_p = \frac{\angle\phi_p}{2\pi} \qquad p = 1, 2, \ldots, P \tag{20}$$

where $\phi_p$s are $\Psi$ eigenvalues and $\angle\phi_p$s are $\phi_p$ phases. Figure 2 depicts the ESPRIT process in the estimation of the signal parameters.

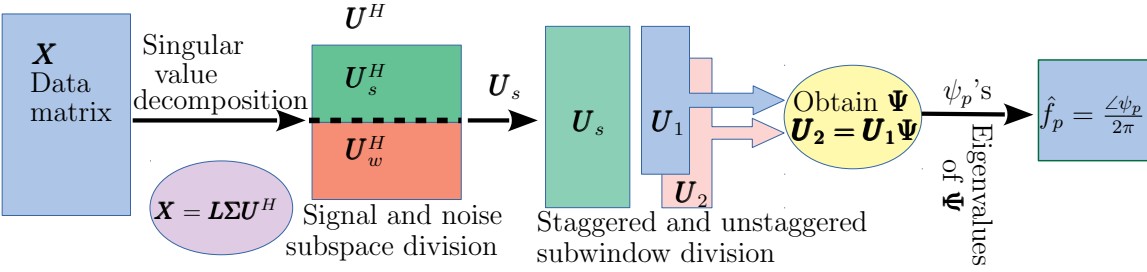

**Figure 2.** Demonstrating the block diagram of the ESPRIT algorithm.

### 3.2. FB-ESPRIT

FB-ESPRIT is indeed the decomposition of the signal to its frequency sub-bands, then individually applying ESPRIT to each sub-band. Since at each resultant sub-band, the content of the other sub-bands in its spectrum has been already removed by the filter bank, thus, prior to individual application of ESPRIT to the corresponding sub-band, its spectrum is spread to cover all the spectrum. The dedication of the full bandwidth to one of the $L$ sub-bands brings multiple times greater efficiency and accuracy to parameter estimation.

The step-by-step process of the FB-ESPRIT algorithm is as follows:

Step 1: The signal $s(t)$ goes through a uniform filter bank of $L$ filters, and the signal is decomposed into $L$ sub-bands $s_0(t), s_1(t), \ldots, s_L(t)$.

Step 2: The spectrum of each sub-band signals $s_i$ is spread $L$ times through down-sampling of factor $L$.

Step 3: Each resulting down-sampled sub-band signal $s_i^{L\downarrow}$ goes through a parameter estimation by ESPRIT.

Step 4: The down-sampling frequency shifting effect is removed from the estimated frequencies by a relocation process.

Figure 3 depicts the FB-ESPRIT process.

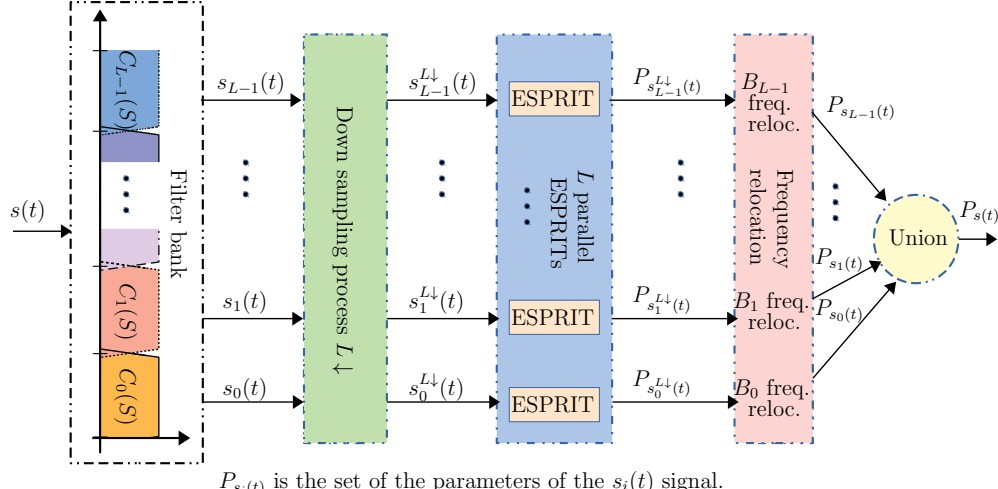

$P_{s_i(t)}$ is the set of the parameters of the $s_i(t)$ signal.

**Figure 3.** The filter bank associated ESPRIT block diagram.

### 3.3. FB-ESPRIT vs. Colored Noise

This section illustrates why FB-ESPRIT should be less affected by colored noise. It is indeed due to the effect of the filter bank and downsampling inherent to FB-ESPRIT, as explained below. In order to give a clear view, let us analyze and compare the colored noise spectrum after the filter bank and downsampling before going through an internal ESPRIT block of FB-ESPRIT. The colored noise has a spectrum shape that is not monotonic like the spectrum of white noise. ESPRIT's performance is dependent on the whiteness of the noise, and because of that, the colored noise degrades its efficiency. In FB-ESPRIT, the non-monotonic spectrum of the colored noise is decomposed into sub-bands by the filter bank implementation. Then, each resultant sub-band is down-sampled prior to the ESPRIT process. Even though the spectrum of colored noise is non-monotonic, along a sub-band of the noise, it can be more monotonic than the general spectrum. As the sub-band is down-sampled, its spectrum is spread over the full bandwidth of the noise, and it is flattened even more. Therefore, the filter bank and down-sampling make each sub-band spectrum closer to a white monotonic shape, and the corresponding individual ESPRIT can perform more efficiently. Figure 4 depicts the above mentioned effect by the filter bank and down-sampling on pink noise where two of the sub-bands get a more monotonic spectrum shape after the filter bank and down-sampling process compared to the full-spectrum in the case of non-filtered Pink noise.

On the other hand, apart from the noise spectrum shape, which becomes more monotonic, the energy of the noise is divided amongst the sub-bands. Therefore, each spread spectrum sub-band has $1/L$ times the initial noise power.

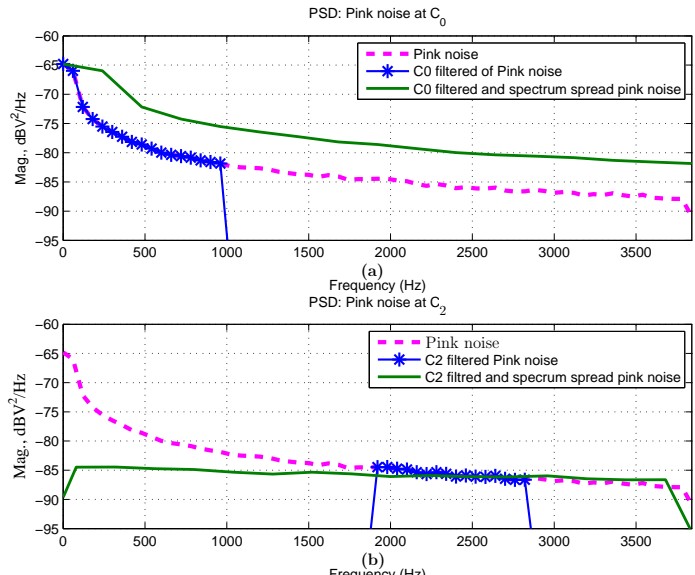

**Figure 4.** The stretching effect of the filter bank (FB) followed by spectrum spread inherent to FB-ESPRIT on the pink noise sub-band spectrum; (**a**) the subband $C_0$, and (**b**) the subband $C_2$ after spread spectrum effect.

## 4. Result Analysis and Discussion

Normally, a power line signal contains a fundamental component of 60 Hz and some harmonics. To have a comprehensive evaluation of the colored noise effect on the estimation of the signal parameters of frequency, amplitude, and phase for all the signal components, we performed the evaluative simulations over two synthesized power signals containing a sub-harmonic, multiple harmonics, and interharmonics in addition to the fundamental component. The synthetic power signals represent a complex set of parameters where each is differently affected by colored noise. The analytical comparison was between the performance of FB-ESPRIT and the conventional ESPRIT in the estimation of the parameters of the harmonics, interharmonics, and subharmonics of the synthesized power signals under different colored noise conditions. To have a clear analysis of the advantages of FB-ESPRIT, for this paper, the filters' coefficients were obtained through the Johnston optimized filter design [47]. The base of the Johnston 12 dB filter was used to compose the structure design of filters [48].

Thus, we acquired $L = 4$. Therefore, the filter bank divided the signal spectrum into four sub-bands with an equal bandwidth by four filters as (i) $C_0(z)$ a low pass filter with a cutoff frequency of 960 Hz, (ii) $C_1(z)$ a band pass filter with low and high cutoff frequencies at 960 Hz and 1920 Hz, (iii) another band pass filter with low and high cutoff frequencies at 1920 and 2880 Hz, and (iv) a high pass filter with a cutoff frequency at 2880 Hz.

### 4.1. FB-ESPRIT vs. ESPRIT in Power Quality Analysis under the Colored Noisy Condition

As a power signal with a complex content of components, a synthesized sinusoidal signal with multiple harmonics, interharmonics, and sub-harmonic components was acquired as the case study of parameter estimation for evaluation and comparison of FB-ESPRIT with conventional ESPRIT in the presence of colored noise. The synthesized signal $s_1(t)$ is as follows:

$$
\begin{aligned}
s_1(t) = \cos(2\pi 60 t + 15^\circ) \; &+ \; 0.2\cos(2\pi 30 t + 25^\circ) + 0.3\cos(2\pi 27.5 \times 60 t + 45^\circ) \\
&+ 0.2\cos(2\pi 43.5 \times 60 t + 60^\circ) + 0.1\cos(2\pi 55 \times 60 t + 68^\circ).
\end{aligned}
\tag{21}
$$

Signal $s_1(t)$ in Equation (21) contains the fundamental component, two interharmonics, one sub-harmonic, and a harmonic. The parameters aimed to be estimated were the frequencies, amplitudes, and phases of the sinusoidal components.

The evaluative analysis of the estimation of $s_1(t)$ parameters was performed through 1000 runs for each sample with SNRs of 5, 20, 30, and 40 dB for the signal $s_1(t)$ contaminated by pink, red, blue, and violet noises. The number of points per cycle $N_{ppc}$ for FB-ESPRIT and ESPRIT was acquired as 32 points, as they were implemented over two cycles of the signal duration. The acquired sampling frequency was 7680 Hz, which guaranteed the ability to measure frequencies up to 3840 Hz. Figures 5 and 6 show the power spectral density (PSD) of pink, red, blue, and violet additive noise to the signal with an SNR of approximately 5 dB.

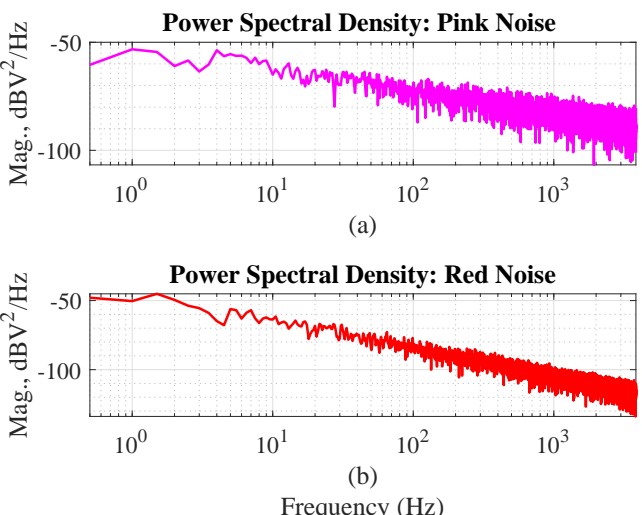

**Figure 5.** Power spectral density with of the additive (**a**) pink noise and (**b**) red noise of $SNR = 5$ dB.

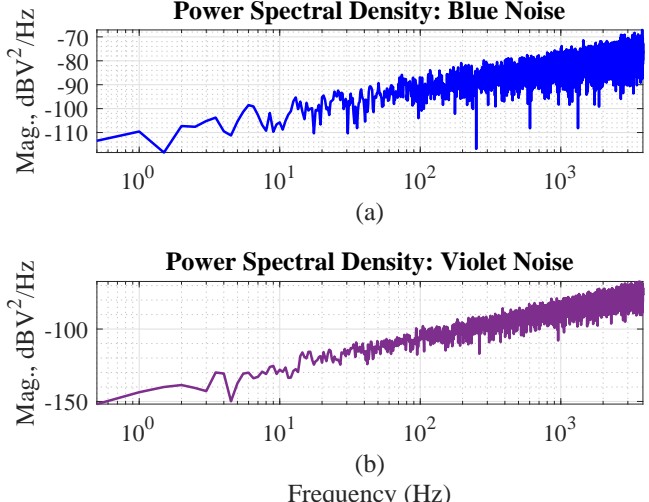

**Figure 6.** Power spectral density with of the additive (**a**) blue noise and (**b**) violet noise of $SNR = 5$ dB.

Tables 2 and 3 respectively indicate the mean of the estimated parameters and the percentage error for the signal $s_1(t)$ in 1000 runs at each signal-to-noise ratio (SNR) of 5, 20, 30, and 40 dB considering $s_1(t)$ contaminated by pink noise.

**Table 2.** Mean of $s_1(t)$ signal parameters estimated in 1000 runs for the FB-ESPRIT method and the conventional ESPRIT at different SNR levels (5, 20, 30, 40 dB) of pink noise.

| SNR | 5 dB | | 20 dB | | 30 dB | | 40 dB | |
|---|---|---|---|---|---|---|---|---|
| **Estimator** | **FB-ESP.** | **ESPRIT** | **FB-ESP.** | **ESPRIT** | **FB-ESP.** | **ESPRIT** | **FB-ESP.** | **ESPRIT** |
| Fund. F. : 60 Hz | 60.02 | 52.80 | 59.96 | 60.00 | 59.99 | 60.00 | 60.00 | 60.00 |
| Int. F. : 30 Hz | 26.71 | 87.70 | 27.87 | 31.10 | 29.56 | 30.20 | 29.97 | 30.00 |
| Har. F.: 1650 | 1649.90 | 1649.00 | 1650.00 | 1650.00 | 1650.00 | 1650.00 | 1650.00 | 1650.00 |
| Har. F.: 2610 | 2610.60 | 2608.90 | 2610.00 | 2610.00 | 2610.00 | 2610.00 | 2610.00 | 2610.00 |
| Har. F.: 3300 | 3299.70 | 227.10 | 3300.00 | 3300.00 | 3300.00 | 3300.00 | 3300.00 | 3300.00 |
| Fund. 60 A.: 1 | 1.06 | 0.70 | 0.96 | 0.91 | 0.99 | 1.10 | 1.01 | 1.07 |
| Int. 30 A.: 0.2 | 0.18 | 0.08 | 0.20 | 0.34 | 0.20 | 0.40 | 0.20 | 0.21 |
| Har. 1650 A.: 0.3 | 0.29 | 0.05 | 0.29 | 0.30 | 0.29 | 0.30 | 0.29 | 0.30 |
| Har. 2610 A.: 0.2 | 0.20 | 0.16 | 0.20 | 0.21 | 0.20 | 0.20 | 0.20 | 0.20 |
| Har. 3300 A.: 0.1 | 0.09 | 0.08 | 0.10 | 0.11 | 0.10 | 0.10 | 0.10 | 0.10 |
| Fund. 60 $\phi$: 15° | 15.09 | 1.68 | 20.15 | 3.95 | 15.28 | 8.58 | 15.20 | 13.72 |
| Sub. 30 $\phi$: 25° | 23.95 | 26.91 | 26.73 | 52.20 | 27.32 | 39.18 | 26.77 | 0,10 |
| Int. 1650 $\phi$: 45° | 43.05 | 53.06 | 49.90 | 81.98 | 46.96 | 47.00 | 44.88 | 45.21 |
| Har. 2610 $\phi$: 60° | 57.14 | 55.79 | 64.61 | 61.17 | 56.54 | 61.04 | 61.61 | 60.12 |
| Har. 3300 $\phi$: 68° | 68.47 | 86.77 | 68.80 | 66.44 | 69.12 | 70.04 | 67.92 | 68.24 |

**Table 3.** Percentage error (%) of signal $s_1(t)$ parameters' estimated with 1000 runs for the FB-ESPRIT method and the conventional ESPRIT method at different SNR levels (5, 20, 30, 40 dB) of pink noise.

| SNR | 5 dB | | 20 dB | | 30 dB | | 40 dB | |
|---|---|---|---|---|---|---|---|---|
| **Estimator** | **FB-ESP.** | **ESPRIT** | **FB-ESP.** | **ESPRIT** | **FB-ESP.** | **ESPRIT** | **FB-ESP.** | **ESPRIT** |
| Fund. F: 60 Hz | 0.03 | 12.00 | 0.06 | 0.00 | 0.01 | 0.00 | 0.00 | 0.00 |
| Int. F.: 30 Hz | 10.97 | 99.00 | 7.09 | 3.67 | 1.45 | 0.67 | 0.09 | 0.00 |
| Har. F.: 1650 | 0.01 | 0.06 | 0.00 | 0.00 | 0.00 | 0.00 | 0.00 | 0.00 |
| Har. F.: 2610 | 0.02 | 0.04 | 0.00 | 0.00 | 0.00 | 0.00 | 0.00 | 0.00 |
| Har. F.: 3300 | 0.01 | 93.12 | 0.00 | 0.00 | 0.00 | 0.00 | 0.00 | 0.00 |
| midrule Fund. 60 A.: 1 | 5.57 | 29.94 | 4.31 | 8.86 | 0.52 | 6.96 | 0.59 | 7.40 |
| Int. 30 A.: 0.2 | 12.40 | 58.00 | 1.15 | 70.95 | 0.45 | 97.75 | 0.75 | 6.50 |
| Har. 1650 A.: 0.3 | 3.70 | 83.10 | 2.87 | 1.60 | 2.17 | 0.83 | 2.77 | 0.43 |
| Har. 2610 A.: 0.2 | 1.80 | 20.95 | 0.45 | 5.00 | 1.15 | 0.15 | 0.20 | 2.60 |
| Har. 3300 A.: 0.1 | 7.70 | 15.20 | 1.10 | 7.70 | 1.10 | 1.30 | 0.60 | 0.70 |
| Fund. 60 $\phi$: 15° | 0.58 | 88.81 | 34.31 | 73.65 | 1.89 | 42.77 | 1.31 | 8.56 |
| Sub. 30 $\phi$: 25° | 4.18 | 7.62 | 6.91 | 96.78 | 9.29 | 56.73 | 7.09 | 99.60 |
| Int. 1650 $\phi$: 45° | 4.34 | 17.91 | 10.89 | 82.18 | 4.35 | 4.44 | 0.,27 | 0.47 |
| Har. 2610 $\phi$: 60° | 4.77 | 7.01 | 7.68 | 1.96 | 5.76 | 1.73 | 2.69 | 0.19 |
| Har. 3300 $\phi$: 68° | 0.69 | 27.60 | 1.17 | 2.30 | 1.65 | 3.01 | 0.12 | 0.35 |

Tables 4–6 respectively indicate the percentage error for signal $s_1(t)$ for the 1000 runs for each SNR level of 5, 20, 30, and 40 dB of red, blue, and violet noises. Figures 7 and 8 compare the robustness of FB-ESPRIT and ESPRIT by means of the statistical deviations of the estimated fundamental frequencies of $s_1(t)$ in 1000 runs at each SNR level of 5, 20, 30, and 40 dB of different colored noises. Figures 9–11 visually compare the robustness of ESPRIT and FB-ESPRIT by means of the statistical deviations of the estimated frequency, amplitude, and phase errors of the fundamental component of $s_1(t)$ in 1000 runs at each SNR level of 5, 20, 30, and 40 dB of colored noises.

**Table 4.** Percent error (%) in signal $s_1(t)$ parameters' estimation in 1000 runs by FB-ESPRIT and the conventional ESPRIT method at different SNR levels (5, 20, 30, 40 dB) under contamination of red noise.

| SNR | 5 dB | | 20 dB | | 30 dB | | 40 dB | |
|---|---|---|---|---|---|---|---|---|
| **Estimator** | **FB-ESP.** | **ESPRIT** | **FB-ESP.** | **ESPRIT** | **FB-ESP.** | **ESPRIT** | **FB-ESP.** | **ESPRIT** |
| Fund. F: 60 Hz | 0.32 | 3.50 | 0.09 | 0.00 | 0.02 | 0.00 | 0.00 | 0.00 |
| Int. F. : 30 Hz | 16.55 | 22.33 | 18.93 | 14.67 | 3.22 | 1.33 | 0.32 | 0.33 |
| Har. F.: 1650 | 0.00 | 0.01 | 0.00 | 0.00 | 0.00 | 0.00 | 0.00 | 0.00 |
| Har. F.: 2610 | 0.00 | 0.01 | 0.00 | 0.00 | 0.00 | 0.00 | 0.00 | 0.00 |
| Har. F.: 3300 | 0.00 | 81.18 | 0.00 | 0.00 | 0.00 | 0.00 | 0.00 | 0.00 |

**Table 4.** *Cont.*

| SNR | 5 dB | | 20 dB | | 30 dB | | 40 dB | |
|---|---|---|---|---|---|---|---|---|
| Estimator | FB-ESP. | ESPRIT | FB-ESP. | ESPRIT | FB-ESP. | ESPRIT | FB-ESP. | ESPRIT |
| Fund. 60 A.: 1 | 0.62 | 62.29 | 1.21 | 7.40 | 0.63 | 1.23 | 0.09 | 0.17 |
| Int. 30 A.: 0.2 | 19.20 | 82.80 | 0.25 | 78.95 | 0.70 | 11.15 | 3.40 | 5.00 |
| Har. 1650 A.: 0.3 | 2.37 | 1.30 | 2.87 | 0.17 | 2.73 | 0.03 | 2.73 | 0.03 |
| Har. 2610 A.: 0.2 | 0.50 | 0.55 | 0.10 | 0.20 | 0.15 | 0.10 | 0.15 | 0.15 |
| Har. 3300 A.: 0.1 | 8.70 | 46.00 | 0.80 | 36.90 | 0.80 | 0.10 | 0.70 | 0.00 |
| Fund. 60 $\phi$: 15° | 15.91 | 96.05 | 31.83 | 50.83 | 9.68 | 3.23 | 0.10 | 3.11 |
| Sub. 30 $\phi$: 25° | 38.16 | 100.40 | 4.63 | 98.35 | 1.86 | 96.46 | 0.02 | 49.58 |
| Int. 1650 $\phi$: 45° | 5.56 | 46.01 | 0.67 | 0.94 | 0.80 | 0.19 | 0.84 | 0.03 |
| Har. 2610 $\phi$: 60° | 5.44 | 4.07 | 2.03 | 0.65 | 0.15 | 0.12 | 0.31 | 0.02 |
| Har. 3300 $\phi$: 68° | 1.13 | 0.27 | 0.29 | 2.05 | 0.02 | 0.13 | 0.03 | 0.02 |

**Table 5.** Percent error (%) in signal $s_1(t)$ parameters' estimation in 1000 runs by FB-ESPRIT and the conventional ESPRIT method at different SNR levels (5, 20, 30, 40 dB) under contamination of blue noise.

| SNR | 5 dB | | 20 dB | | 30 dB | | 40 dB | |
|---|---|---|---|---|---|---|---|---|
| Estimator | FB-ESP. | ESPRIT | FB-ESP. | ESPRIT | FB-ESP. | ESPRIT | FB-ESP. | ESPRIT |
| Fund. F.: 60 Hz | 0.17 | 2.00 | 0.00 | 0.50 | 0.00 | 0.00 | 0.00 | 0.00 |
| Int. F. : 30 Hz | 49.53 | NA | 0.01 | NA | 0.00 | 2.00 | 0.00 | 0.00 |
| Har. F.: 1650 | 0.00 | 0.08 | 0.00 | 0.01 | 0.00 | 0.00 | 0.00 | 0.00 |
| Har. F.: 2610 | 0.00 | 0.11 | 0.00 | 0.00 | 0.00 | 0.00 | 0.00 | 0.00 |
| Har. F.: 3300 | 0.00 | 0.43 | 0.00 | 0.01 | 0.00 | 0.00 | 0.00 | 0.00 |
| Fund. 60 A.: 1 | 1.91 | 90.09 | 0.13 | 8.14 | 0.06 | 15.42 | 0.00 | 2.00 |
| Int. 30 A.: 0.2 | 19.25 | NA | 0.65 | NA | 0.25 | 96.75 | 0.05 | 3.15 |
| Har. 1650 A.: 0.3 | 3.57 | 88.60 | 1.50 | 1.97 | 2.80 | 1.13 | 2.83 | 0.50 |
| Har. 2610 A.: 0.2 | 0.20 | 83.40 | 4.30 | 4.35 | 0.55 | 3.90 | 0,70 | 0.75 |
| Har. 3300 A.: 0.1 | 0.80 | 63.10 | 10.80 | 20.30 | 0.40 | 3.70 | 0.30 | 0.20 |
| Fund. 60 $\phi$: 15° | 4.32 | 98.57 | 0.30 | 96.13 | 0.10 | 13.72 | 0.02 | 0.77 |
| Sub. 30 $\phi$: 25° | 22.00 | NA | 1.32 | NA | 0.53 | 28.29 | 0.16 | 28.21 |
| Int. 1650 $\phi$: 45° | 1.52 | 85.83 | 18.81 | 10.31 | 6.80 | 3.67 | 1.79 | 0.30 |
| Har. 2610 $\phi$: 60° | 0.35 | 33.10 | 17.43 | 0.54 | 2.13 | 4.36 | 0.12 | 0.39 |
| Har. 3300 $\phi$: 68° | 0.84 | 32.20 | 2.34 | 40.64 | 1.13 | 0.33 | 0.22 | 1.28 |

**Table 6.** Percent error (%) of signal $s_1(t)$ parameters estimated in 1000 runs of the FB-ESPRIT method and the conventional ESPRIT method at different SNR levels (5, 20, 30, 40 dB) while the signal is contaminated with violet noise.

| SNR | 5 dB | | 20 dB | | 30 dB | | 40 dB | |
|---|---|---|---|---|---|---|---|---|
| Estimator | FB-ESP. | ESPRIT | FB-ESP. | ESPRIT | FB-ESP. | ESPRIT | FB-ESP. | ESPRIT |
| Fund. F.: 60 Hz | 0.00 | 2.00 | 0.00 | 0.50 | 0.00 | 0.17 | 0.00 | 0.00 |
| Int. F. : 30 Hz | 0.06 | NA | 0.01 | NA | 0.00 | 11.33 | 0.00 | 0.33 |
| Har. F.: 1650 | 0.01 | 0.14 | 0.00 | 0.01 | 0.00 | 0.00 | 0.00 | 0.00 |
| Har. F.: 2610 | 0.11 | 0.22 | 0.00 | 0.01 | 0.00 | 0.00 | 0.00 | 0.00 |
| Har. F.: 3300 | 0.24 | 1.05 | 0.00 | 0.02 | 0.00 | 0.00 | 0.00 | 0.00 |
| Fund. 60 A.: 1 | 0.44 | 83.96 | 0.01 | 6.96 | 0.02 | 31.43 | 0.03 | 0.64 |
| Int. 30 A.: 0.2 | 4.25 | NA | 0.01 | NA | 0.10 | 95.00 | 0.10 | 14.55 |
| Har. 1650 A.: 0.3 | 6.40 | 70.00 | 4.70 | 5.10 | 2.10 | 0.10 | 2.80 | 0.30 |
| Har. 2610 A.: 0.2 | 17.65 | 81.90 | 9.60 | 12.55 | 0.60 | 2.70 | 0.30 | 0.15 |
| Har. 3300 A.: 0.1 | 34.90 | 68.50 | 17.10 | 41.20 | 1.00 | 5.30 | 0.80 | 2.20 |
| Fund. 60 $\phi$: 15° | 1.11 | 80.21 | 0.02 | 12.37 | 0.09 | 92.63 | 0.04 | 1.59 |
| Sub. 30 $\phi$: 25° | 4.68 | NA | 1.88 | NA | 0.29 | 0.39 | 0.12 | 32.64 |
| Int. 1650 $\phi$: 45° | 12.89 | 20.71 | 6.39 | 27.65 | 0.23 | 4.63 | 0.01 | 0.17 |
| Har. 2610 $\phi$: 60° | 3.12 | 8.59 | 0.53 | 1.39 | 0.39 | 3.91 | 0.98 | 0.33 |
| Har. 3300 $\phi$: 68° | 3.99 | 6.88 | 1.98 | 59.39 | 0.20 | 63.20 | 0.35 | 2.28 |

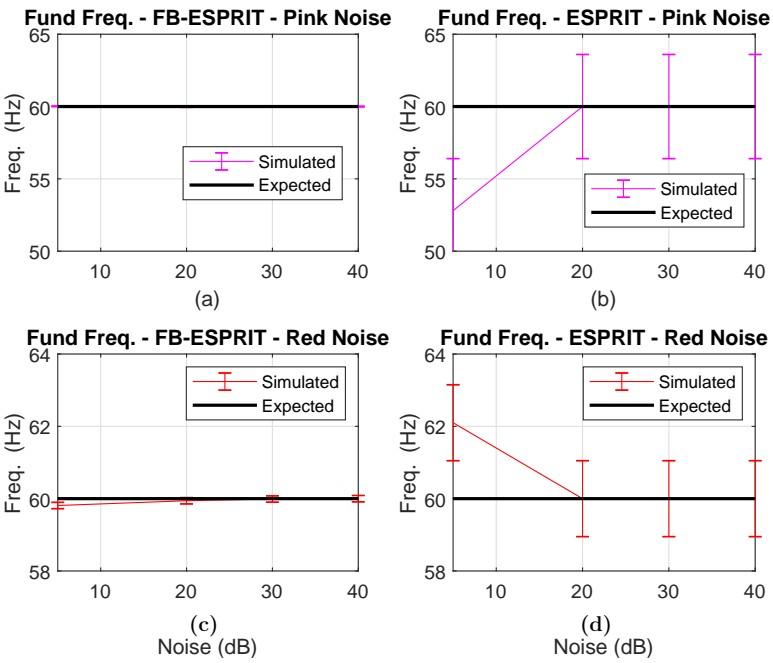

**Figure 7.** Estimation of fundamental frequency (Fund Freq.) when $s_1(t)$ is contaminated with pink and red noises; (**a**) by FB-ESPRIT under pink noise, (**b**) by ESPRIT under pink noise, (**c**) by FB-ESPRIT under red noise, and (**d**) by ESPRIT under red noise.

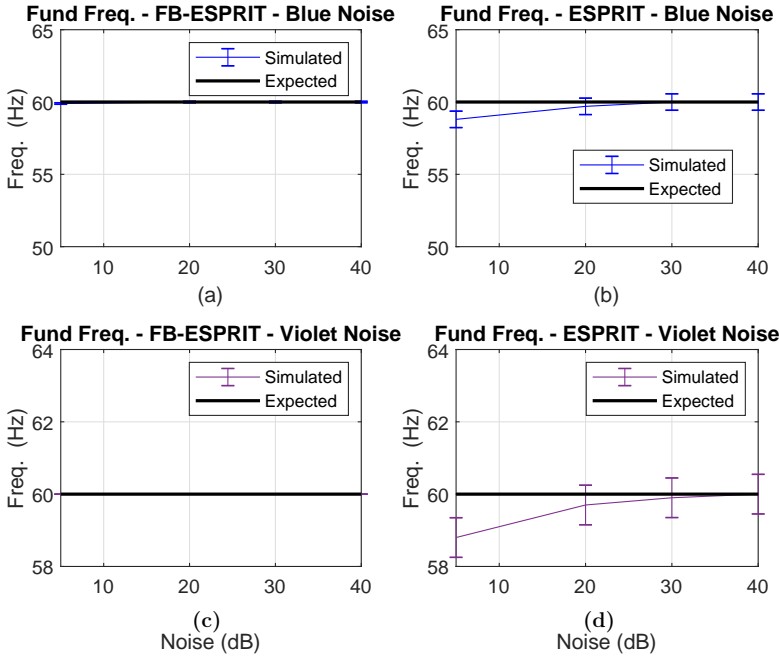

**Figure 8.** Estimation of fundamental frequency (Fund Freq.) when $s_1(t)$ is contaminated with blue and violet noises; (**a**) by FB-ESPRIT under blue noise, (**b**) by ESPRIT under blue noise, (**c**) by FB-ESPRIT under violet noise, and (**d**) by ESPRIT under violet noise.

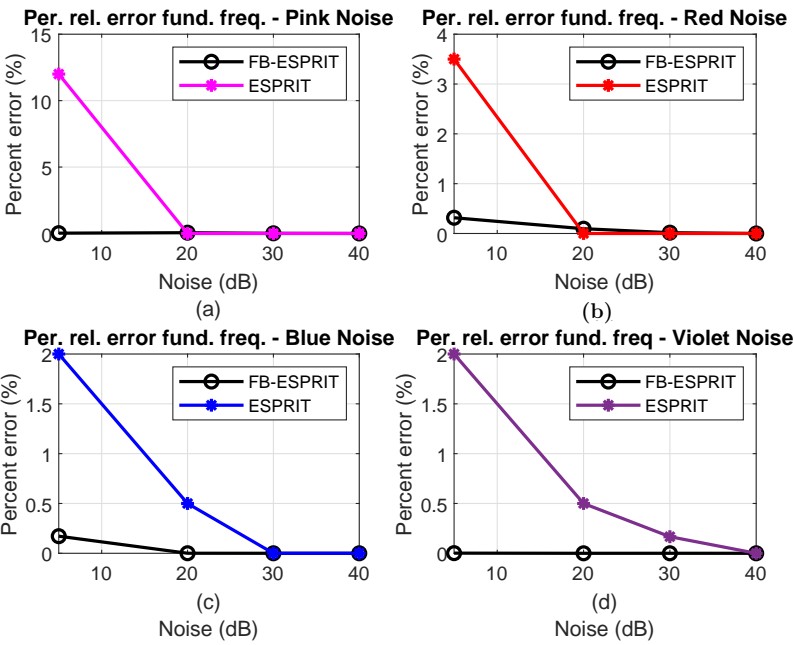

**Figure 9.** Percentage of relative (Per. rel.) error (%) of 60 Hz fundamental frequency (fund. freq.) estimation: (**a**) signal contaminated with pink noise, (**b**) signal contaminated with red noise, (**c**) signal contaminated with blue noise, and (**d**) signal contaminated with violet noise.

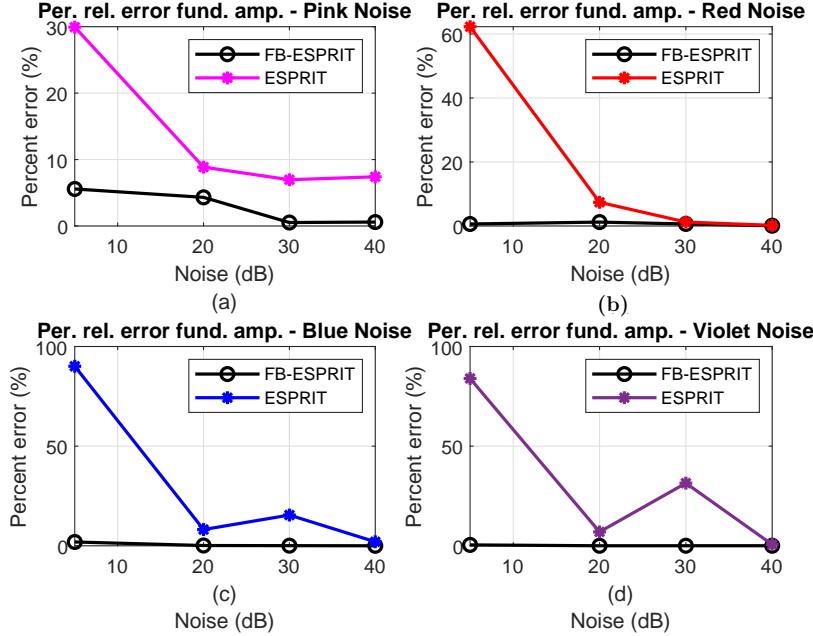

**Figure 10.** Percentage of relative (Per. rel.) error (%) of 60 Hz fundamental component amplitude (fund. amp.): (**a**) signal contaminated with pink noise; (**b**) signal contaminated with red noise; (**c**) signal contaminated with blue noise; (**d**) signal contaminated with violet noise.

Figure 12 visually compares the robustness of ESPRIT and FB-ESPRIT by means of the statistical deviations of the estimated sub-harmonic frequency of 30 Hz for $s_1(t)$ in 1000 runs at each SNR level of

5, 20, 30, and 40 dB while the $s_1(t)$ is contaminated by pink and red noises. The data vector used for the autocorrelation in ESPRIT and FB-ESPRIT had a length of 128 and 32 samples, respectively. This means FB-ESPRIT was set to fast performance, but equal efficiency to ESPRIT under standard conditions.

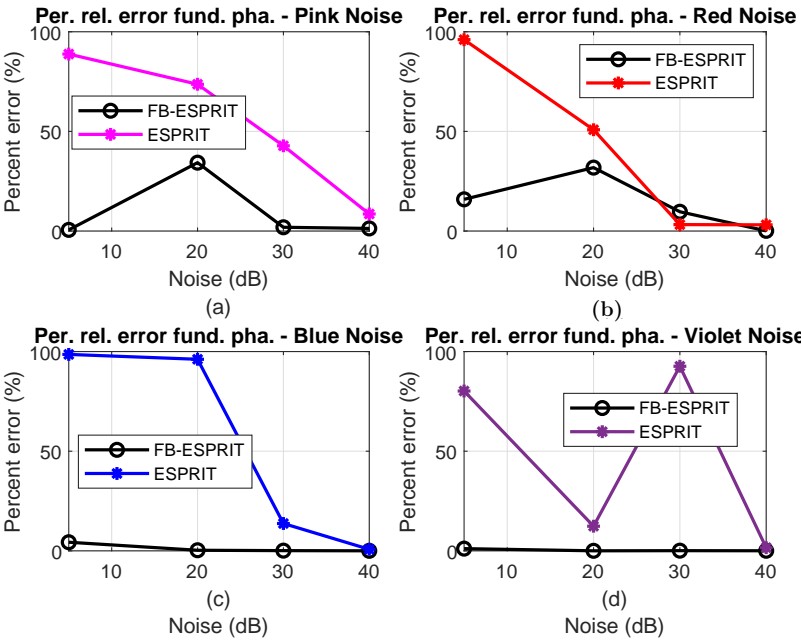

**Figure 11.** Percentage relative (Per. rel.) error (%) of 60 Hz fundamental component phase: (**a**) signal contaminated with pink noise, (**b**) signal contaminated with red noise, (**c**) signal contaminated with blue noise, (**d**) signal contaminated with violet noise.

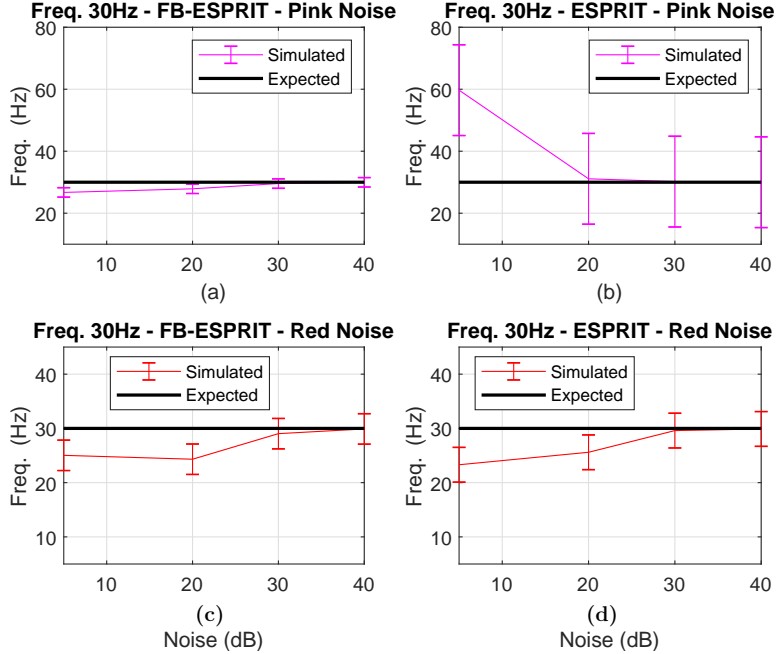

**Figure 12.** Estimation of the 30 Hz sub-harmonic frequency of the $s_1(t)$ signal contaminated with pink and red noises, indicating statistical deviations for *SNR* variation from 5 to 40 dB: (**a**) by FB-ESPRIT under pink noise, (**b**) by ESPRIT under pink noise, (**c**) by FB-ESPRIT under red noise, and (**d**) by ESPRIT under red noise.

The analysis of the results of the simulations in Tables 2–6 is presented in the following subsections.

### 4.1.1. 30 Hz Sub-Harmonic Parameters Estimation

For signal $s_1(t)$ at the same signal length, while FB-ESPRIT detects the 30 Hz sub-harmonic frequency at 5 and 20 dB for all colored noises, ESPRIT detects this frequency only for pink and red noise with errors of approximately (99.00–3.67%) and (22.33–14.67%), respectively; see Tables 3–6.

As said earlier, for a low signal-to-noise ratio, the pink and red noise signals have higher energy and can influence measurements. For blue and violet noises, even though the relationship is the inverse, the noise level is high ($SNR = 5$ dB), and the 30 Hz sub-harmonic will be the component that will be most influenced by this noise level. However, it is apparent that the FB-ESPRIT method was considerably less influenced by these noises and stood out even amid a high noise level (Tables 3–6).

Analyzing the estimated amplitudes of this component, FB-ESPRIT indicated its worst case for blue noise with an error of 19.25% at 5 dB (Table 5), but at the same noise level, the ESPRIT technique did not even estimate the parameters. Considerable errors by the ESPRIT technique were observed under different SNR levels of colored noises (Tables 3–6).

In the evaluation for phase estimation, the 30 Hz subharmonic, the BF-ESPRIT indicated its worst case for the red noise at 5 dB with an error of approximately (38.16%) (Table 4), but at this same noise level, the ESPRIT technique presented an error above 100%. Moreover, the ESPRIT technique obtained considerably high errors; under pink noise, estimation of the 30 Hz component had errors around $(17.91, 82.18, 4.44, 0.47\%)$ approximately (Table 3), under red noise $(100.40–98.35–96.46–49.58\%)$ (Table 4), under blue noise (NA, NA, $28.29, 28.21\%$) (Table 5), and under violet noise (NA, NA, $0.39, 32.64$).

Thus, in the 30 Hz component estimation in practical analysis, FB-ESPRIT indicated a superior estimation of frequency, amplitude, and phase parameters under different colored noises. From Table 3, in contamination with pink noise, it is observed that the error measured in 5 dB for the frequency parameter was about nine times smaller for FB-ESPRIT than ESPRIT. The amplitude parameter was about 4.68 times smaller, and the phase was about 4.28 times smaller. In comparison, under other noises, there was also a higher ratio of FB-ESPRIT compared to ESPRIT. Figure 12 shows the 30 Hz sub-harmonic frequency estimates for the signal $s_1(t)$ contaminated with pink and red noises, indicating statistical deviations and showing how ESPRIT deviated more from the expected value than FB-ESPRIT.

### 4.1.2. 60 Hz Fundamental Parameters' Estimation

Evaluating the 60 Hz fundamental component frequency estimate, it was observed that the FB-ESPRIT method performed in all situations, and its worst case error was under a 5 dB red noise error of approximately 0.32% (Table 4), while the ESPRIT technique indicated errors of up to 3.5%. Moreover, when the signal $s_1(t)$ was contaminated with pink noise, the ESPRIT at 5dB had errors of up to 12.00% (Table 3). Observe Figures 7 and 8, which display the 60 Hz fundamental frequency estimates for the signal $s_1(t)$ contaminated with colored noises. Note in these figures the statistical deviations of the ESPRIT technique as it deviates more than the expected value compared to FB-ESPRIT. Note also Figure 9, which displays the fundamental frequency percentage error for the signal $s_1(t)$ contaminated with colored noises according to 1000 runs. Figure 9 also indicates the higher efficiency of FB-ESPRIT compared to ESPRIT.

Analyzing the amplitudes' estimations, ESPRIT presented a more critical situation in the estimations. While the FB-ESPRIT method indicated a maximum error of 5.57% under pink noise of 5 dB, the ESPRIT technique showed an error of 29.94% (Table 3). In addition, when the signal $s_1(t)$ was contaminated with blue, violet, and red noises of 5dB, ESPRIT had errors of respectively $90.09, 83.96$ and 62.29% (Tables 4–6). Note that Figure 10 demonstrates the fundamental amplitude percentage error for the signal $s_1(t)$ contaminated with colored noises in 1000 runs. Figure 10 again shows a better FB-ESPRIT calculation compared to the ESPRIT technique.

In the estimation of the phase, when the signal $s_1(t)$ was contaminated with colored noises, the ESPRIT technique showed considerably high errors. Note in Table 4 that the error for this case exceeded 100% by 5 dB and was very close to this value for the other cases (98.35, 96.46%). In addition, when the signal $s_1(t)$ was contaminated with blue and pink noises, the situation repeated, i.e., the errors at 5 and 20 dB were approximately 98.53, 96.13%, and 88.81, 73.65%, respectively. In the case of the FB-ESPRIT method, the worst case was when the signal $s_1(t)$ was contaminated with pink noise at 20 dB, giving an error of approximately 34.31%. Note Figure 11 displaying the fundamental phase percentage error for the signal $s_1(t)$ contaminated with colored noise in 1000 runs. Figure 11 also indicates the better performance of FB-ESPRIT compared to ESPRIT.

Therefore, considering the above-mentioned evaluation for the 60 Hz fundamental component estimation, FB-ESPRIT indicated a superior estimation of frequency, amplitude, and phase parameters for the signal $s_1(t)$ contaminated with colored noise when compared to the technique ESPRIT. It is worth noting that even when the proposed method indicated poor estimates for the phase parameter, it could still be better than the ESPRIT technique.

### 4.1.3. Harmonic 3300 Hz and Inter-Harmonics 1650 Hz and 2610 Hz Parameters' Estimation

In estimating the middle spectrum frequencies, 1650 Hz, and 2610 Hz, including higher noisy conditions, the proposed method had slightly higher efficiency. However, when considering the estimation of the 55[th] harmonic frequency, the ESPRIT technique failed in colored noisy conditions of $SNR = 5$ dB, obtaining errors of approximately 93.12 and 81.18% when the $s_1(t)$ signal was contaminated with pink and red noises, respectively (see Tables 3 and 4).

In estimating the middle frequencies amplitudes, the superiority of FB-ESPRIT was mainly observable in the high SNR of noise of 5 dB. As an example, observe the case of pink noise contamination (see Tables 3 and 4): the percentage error in estimating the amplitude of 1650 Hz was much smaller compared to ESPRIT as $3.70 << 83.10$, as well for the estimated amplitude of 2610 Hz of $1.80 << 20.95$. Similarly considerable dominance of FB-ESPRIT was observable for the cases of blue and violet noises of 5 dB SNR, as we had for the estimated amplitude of 1650 Hz under blue noise $3.57 << 88.60$ and violet noise $6.40 << 70.00$, and in the case of the estimation of a 2610 Hz amplitude under blue and violet noises, we had respectively $0.20 << 83.40$, and $17.65 << 81.90$.

In the estimation of amplitude of the 55[th] harmonic 3300 Hz, the percentage error of the ESPRIT technique was much smaller in the higher noisy condition of the SNR of 5 dB, as we had for pink, blue, red, and violet noises, respectively, $7.70 << 15.20$, $8.70 << 46.00$, $0.80 << 63.10$, and $34.90 << 68.50$ (see Tables 3–6). In the case of higher SNRs, also this dominance was observable.

In the case of the estimation of the middle frequencies phases, the FB-ESPRIT technique had better results on average for 1650 Hz ($SNR = 5$ dB and $SNR = 20$ dB). The best performed cases were for situations where the $s_1(t)$ signal was contaminated with pink and blue noise as $4.34 << 17.91\%$ and $10.89 << 82.18\%$ under pink noise and $0.80 << 63.10$ and $10.80 << 20.30$ under blue noise. When the signal was contaminated with red and violet noise, the FB-ESPRIT technique also indicated high efficiency in an SNR of 5 dB, and its marked ESPRIT showed errors of 46.01% and 20.71%. The FB-ESPRIT method had its worst case when the signal $s_1(t)$ was contaminated with blue noise with an SNR of 20 dB, indicating an error of approximately 18.81%. However, it is clear from Tables 3–6 that even an error above the ESPRIT technique was measured in some cases, but in general, as a matter of fact, FB-ESPRIT estimated with superiority over ESPRIT. For the 55[th] harmonic 3300 Hz phase estimates, the proposed method maintained the superiority to ESPRIT, and its largest measured error occurred when the signal $s_1(t)$ was contaminated with violet noise with $SNR = 5$ as the error percentage was 3.99%. However, the ESPRIT technique for this same situation indicated an error of 6.88%. In addition, at an SNR of 20 dB, the technique indicated an error of 59.39% (Tables 6).

*4.2. Comparison of FB-ESPRIT, ESPRIT, and MUSIC in Power Quality Analysis under Colored Noise*

The synthesized power signal $s_2(t)$ in Equation (22) contained a 60 Hz fundamental, a sub-harmonic, two interharmonics, and two harmonic components. The parameters to be estimated were the frequencies, amplitudes, and phase components of $s_2(t)$. The data vector used for ESPRIT autocorrelation and in each of the ESPRIT-FB internal ESPRITs had a length of 96 and 32 samples for the signal $s_2(t)$, respectively. For the MUSIC algorithm, the data vector used was $M = 96$.

The evaluative analysis of the estimation of the $s_2(t)$ parameters was performed through 1000 executions of $SNR$ variation from 5 to 40 dB, while the number of points per cycle $N_{ppc}$ for the FB-ESPRIT, ESPRIT, and MUSIC algorithms was acquired as 32 points.Tables 7 and 8 indicate the mean of $s_2(t)$ estimated parameters in 1000 runs by FB-ESPRIT, ESPRIT, and MUSIC and the mean percent error the estimations under pink colored noise. Due to the importance of the high level colored noise of 5 dB, the mean percentage error of the estimated parameters by all estimators under pink, red, blue, and violet colored noises are compared in Table 9.

Figures 13–15 indicate the percentage errors of the 60 Hz fundamental component and 1650 Hz and 2650 Hz interharmonic components considering the signal $s_2(t)$ contaminated with pink noise, with the estimates of frequencies, amplitudes, and phases standing out.

Now, for the second case study of power quality analysis under colored noise, we evaluated the FB-ESPRIT, conventional ESPRIT, and MUSIC methods over the $s_2(t)$ synthesized power signal first contaminated with pink noise with SNRs ranging from 5 to 40 dB. Then, the same signal was contaminated with a high level of red, blue, and violet noises of $SNR = 5$ dB. The second study case $s_2(t)$ is as follows:

$$
\begin{aligned}
s_1(t) = {} & \cos(2\pi \times 60t + 25°) + 0.2\cos(2\pi \times 45t + 30°) + \\
& 0.3\cos(2\pi \times 27.5 \times 60t + 45°) + 0.2\cos(2\pi \times 43.5 \times 60t + 60°) + \\
& 0.1\cos(2\pi \times 50 \times 60t + 68°) + 0.1\cos(2\pi \times 63 \times 60t + 50°).
\end{aligned}
\tag{22}
$$

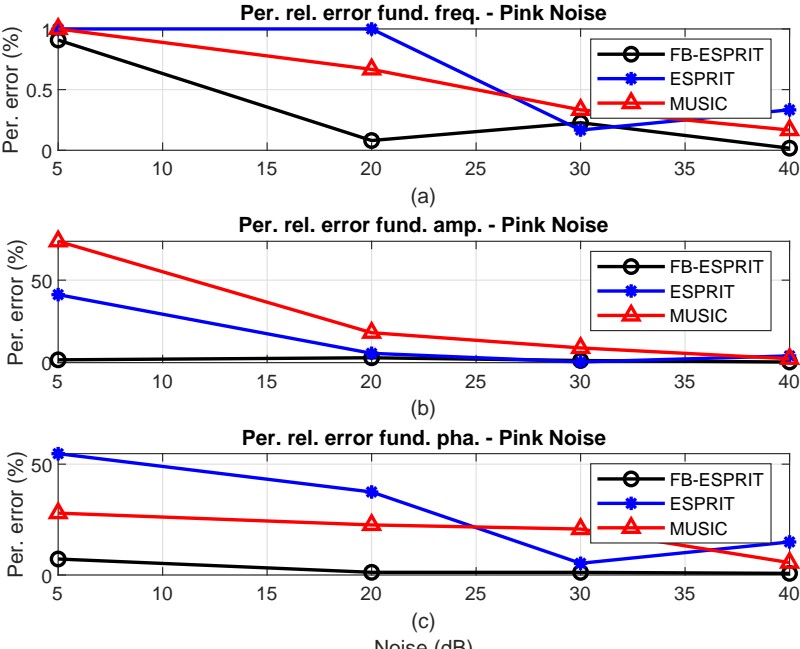

**Figure 13.** Percentage error (%) of the 60 Hz fundamental component when the signal $s_2(t)$ is contaminated with pink noise: (**a**) frequency parameters; (**b**) amplitude parameters; (**c**) phase parameters.

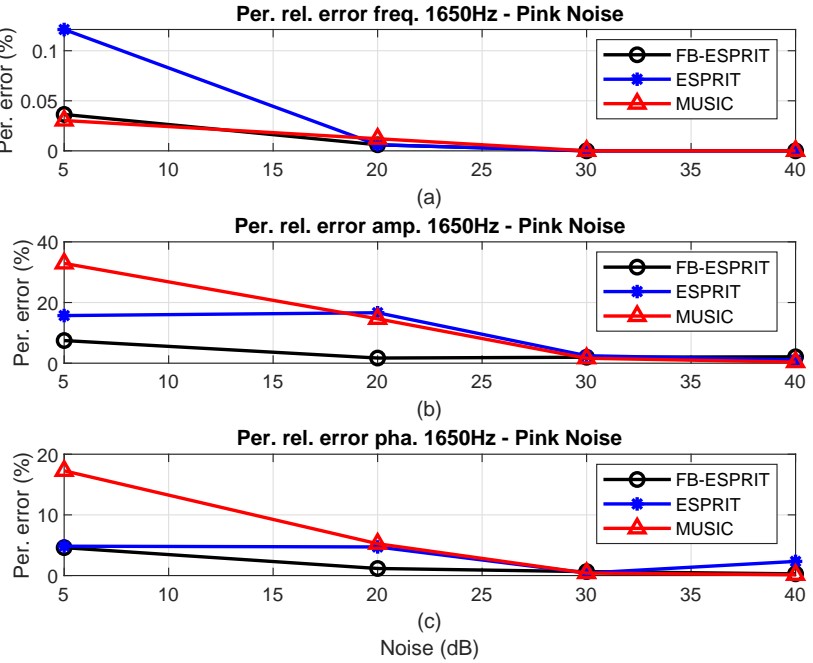

**Figure 14.** Percentage error (%) of the 1650 Hz component when signal $s_2(t)$ is contaminated with pink noise: (**a**) frequency parameters; (**b**) amplitude parameters; (**c**) phase parameters.

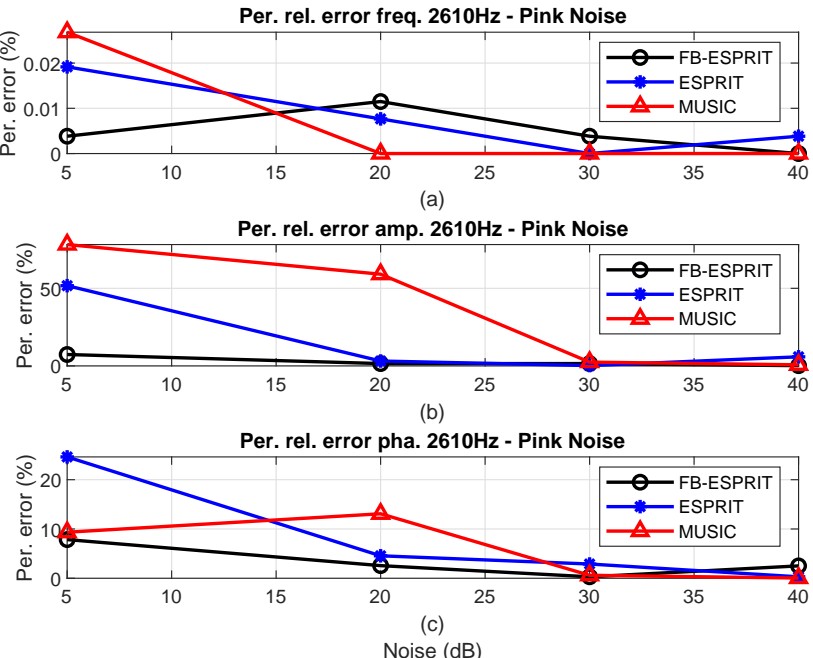

**Figure 15.** Percentage error (%) of the 2610 Hz component when signal $s_2(t)$ is contaminated with pink noise: (**a**) frequency parameters; (**b**) amplitude parameters; (**c**) phase parameters.

**Table 7.** Mean signal $s_2(t)$ parameters estimated in 1000 runs by FB-ESPRIT, ESPRIT, and MUSIC at different levels of SNR (5, 20, 30, 40 dB) under contamination by pink noise.

| SNR | 5 dB | | | 20 dB | | | 30 dB | | | 40 dB | | |
|---|---|---|---|---|---|---|---|---|---|---|---|---|
| **Estimator** | **FB-ESP.** | **ESPRIT** | **MUSIC** | **FB-ESP.** | **ESPRIT** | **MUSIC** | **FB-ESP.** | **ESPRIT** | **MUSIC** | **FB-ESP.** | **ESPRIT** | **MUSIC** |
| Fund. F.: 60 Hz | 60.55 | 59.40 | 59.40 | 59.95 | 60.60 | 59.60 | 59.86 | 60.10 | 59.80 | 60.01 | 60.20 | 59.90 |
| Int. F.: 45 Hz | 36.97 | NA | NA | 44.74 | 43.00 | NA | 44.93 | 43.60 | 46.30 | 44.95 | 40.10 | 45.20 |
| Int. F.: 1650 Hz | 1650.60 | 1648.00 | 1649.50 | 1649.90 | 1649.90 | 1649.80 | 1650.00 | 1650.00 | 1650.00 | 1650.00 | 1650.00 | 1650.00 |
| Int. F.: 2610 | 2610.10 | 2610.50 | 2610.70 | 2610.30 | 2610.20 | 2610.00 | 2609.90 | 2610.00 | 2610.00 | 2610.00 | 2609.90 | 2610.00 |
| Har. F.: 3000 | 3003.10 | NA | NA | 2999.90 | 3000.00 | 3000.20 | 3000.10 | 3000.10 | 3000.00 | 3000.00 | 2999.80 | 3000.00 |
| Har. F.: 3780 | 3779.40 | NA | NA | 3780.10 | 3780.10 | 3779.60 | 3780.00 | 3780.10 | 3779.90 | 3780.00 | 3780.00 | 3780.00 |
| Fund. 60 A.: 1 | 1.02 | 1.41 | 1.74 | 1.03 | 0.94 | 1.18 | 0.99 | 1.00 | 1.09 | 1.00 | 1.04 | 1.02 |
| Int. 45 A.: 0.2 | 0.18 | NA | NA | 0.21 | 0.16 | NA | 0.21 | 0.21 | 0.39 | 0.20 | 0.18 | 0.24 |
| Int. 1650 A.: 0.3 | 0,28 | 0.35 | 0.40 | 0.29 | 0.35 | 0.34 | 0.29 | 0.29 | 0.30 | 0.29 | 0.30 | 0.30 |
| Int. 2610 A.: 0.2 | 0.21 | 0.30 | 0.36 | 0.20 | 0.21 | 0.32 | 0.20 | 0.20 | 0.20 | 0.20 | 0.19 | 0.20 |
| Har. 3000 A.: 0.1 | 0.11 | NA | NA | 0.10 | 0.11 | 0.21 | 0.10 | 0.10 | 0.10 | 0.10 | 0.11 | 0.10 |
| Har. 3780 A.: 0.1 | 0.11 | NA | NA | 0.10 | 0.09 | 0.12 | 0.10 | 0.10 | 0.10 | 0.10 | 0.10 | 0.11 |
| Fund. 60 $\phi$: 25° | 23.20 | 11.31 | 31.98 | 25.30 | 15.64 | 30.65 | 24.70 | 23.68 | 30.18 | 24.82 | 21.27 | 26.39 |
| Int. 45 $\phi$: 30° | 32.10 | NA | NA | 28.00 | 55.70 | NA | 29.42 | 50.56 | 10.21 | 30.48 | 70.56 | 26.92 |
| Int. 1650 $\phi$: 45° | 47.07 | 47.18 | 52.77 | 45.52 | 47.12 | 42.64 | 45.29 | 44.81 | 44.82 | 44.87 | 43.95 | 45.06 |
| Int. 2610 $\phi$: 60° | 64.71 | 74.78 | 54.40 | 61.54 | 62.73 | 52.15 | 59.82 | 58.27 | 60.36 | 61.51 | 59.82 | 59.97 |
| Har. 3000 $\phi$: 68° | 65.48 | NA | NA | 70.13 | 72.25 | 95.80 | 67.86 | 66.74 | 66.20 | 67.99 | 77.56 | 69.06 |
| Har. 3780 $\phi$: 50° | 47.65 | NA | NA | 49.56 | 58.67 | 46.32 | 49.72 | 48.91 | 51.53 | 50.79 | 48.79 | 48.89 |

**Table 8.** Mean percent error (%) of signal $s_2(t)$ parameters estimated in 1000 runs by FB-ESPRIT, ESPRIT, and MUSIC at different levels of SNR (5, 20, 30, 40 dB) under contamination by pink noise.

| SNR | 5 dB | | | 20 dB | | | 30 dB | | | 40 dB | | |
|---|---|---|---|---|---|---|---|---|---|---|---|---|
| Estimator | FB-ESP. | ESPRIT | MUSIC | FB-ESP. | ESPRIT | MUSIC | FB-ESP. | ESPRIT | MUSIC | FB-ESP. | ESPRIT | MUSIC |
| Fund. F.: 60 Hz | 0.91 | 1.00 | 1.00 | 0.08 | 1.00 | 0.67 | 0.23 | 0.17 | 0.33 | 0.02 | 0.33 | 0.17 |
| Int. F.: 45 Hz | 17.85 | NA | NA | 0.57 | 4.44 | NA | 0.15 | 3.11 | 2.89 | 0.11 | 10.89 | 0.44 |
| Int. F.: 1650 Hz | 0.04 | 0.12 | 0.03 | 0.01 | 0.01 | 0.01 | 0.00 | 0.00 | 0.00 | 0.00 | 0.00 | 0.00 |
| Int. F.: 2610 | 0.00 | 0.02 | 0.03 | 0.01 | 0.01 | 0.00 | 0.00 | 0.00 | 0.00 | 0.00 | 0.00 | 0.00 |
| Har. F.: 3000 | 0.10 | NA | NA | 0.00 | 0.00 | 0.01 | 0.00 | 0.00 | 0.00 | 0.00 | 0.01 | 0.00 |
| Har. F.: 3780 | 0.02 | NA | NA | 0.00 | 0.00 | 0.01 | 0.00 | 0.00 | 0.00 | 0.00 | 0.00 | 0.00 |
| Fund. 60 A.: 1 | 1.70 | 41.22 | 73.56 | 2.95 | 5.65 | 18.15 | 1.29 | 0.42 | 8.92 | 0.37 | 4.09 | 2.35 |
| Int. 45 A.: 0.2 | 11.50 | NA | NA | 4.25 | 18.45 | NA | 4.40 | 6.40 | 97.05 | 1.95 | 12.10 | 21.60 |
| Int. 1650 A.: 0.3 | 7.47 | 15.70 | 32.90 | 1.70 | 16.63 | 14.63 | 2.00 | 2.47 | 1.63 | 2.10 | 1.07 | 0.37 |
| Int. 2610 A.: 0.2 | 7.35 | 51.70 | 78.20 | 1.50 | 3.15 | 59.10 | 0.45 | 0.20 | 2.45 | 0.15 | 5.85 | 0.70 |
| Har. 3000 A.: 0.1 | 10.20 | NA | NA | 1.90 | 9.70 | 105.60 | 0.40 | 0.50 | 3.50 | 2.40 | 6.00 | 1.00 |
| Har. 3780 A.: 0.1 | 7.00 | NA | NA | 3.40 | 5.10 | 16.60 | 4.70 | 0.20 | 2.20 | 3.70 | 3.00 | 6.00 |
| Fund. 60 $\phi$: 25° | 7.21 | 54.74 | 27.91 | 1.20 | 37.45 | 22.59 | 1.18 | 5.28 | 20.73 | 0.73 | 14.90 | 5.58 |
| Int. 45 $\phi$: 30° | 6.99 | NA | NA | 6.65 | 85.66 | NA | 192 | 68.54 | 65.98 | 1.61 | 135.21 | 10.26 |
| Int. 1650 $\phi$: 45° | 4.60 | 4.84 | 17.27 | 1.17 | 4.72 | 5.24 | 0.65 | 0.42 | 0.41 | 0.28 | 2.33 | 0.12 |
| Int. 2610 $\phi$: 60° | 7.85 | 24.63 | 9.34 | 2.56 | 4.56 | 13.09 | 0.30 | 2.89 | 0.60 | 2.51 | 0.29 | 0.05 |
| Har. 3000 $\phi$: 68° | 3.70 | NA | NA | 3.13 | 6.25 | 40.88 | 0.20 | 1.85 | 2.65 | 0.01 | 14.06 | 1.55 |
| Har. 3780 $\phi$: 50° | 4.70 | NA | NA | 0.89 | 17.34 | 7.35 | 0.56 | 2.18 | 3.07 | 1.58 | 2.42 | 2.22 |

**Table 9.** Percent error (%) in the estimation of the parameters of signal $s_2(t)$ contaminated by pink, red, blue, and violet noises with an SNR of 5 dB in 1000 runs of FB-ESPRIT, ESPRIT, and MUSIC.

| NOISE | Pink | | | Red | | | Blue | | | Violet | | |
|---|---|---|---|---|---|---|---|---|---|---|---|---|
| SNR | 5 dB | | | 5 dB | | | 5 dB | | | 5 dB | | |
| Estimator | FB-ESP. | ESPRIT | MUSIC | FB-ESP. | ESPRIT | MUSIC | FB-ESP. | ESPRIT | MUSIC | ESP.-BF | ESPRIT | MUSIC |
| Fund. F.: 60 Hz | 0.91 | 1.00 | 1.00 | 0.92 | 0.17 | 5.00 | 1.01 | 1.00 | 5.00 | 0.01 | 1.00 | 5.00 |
| Int. F.: 45 Hz | 17.85 | NA | NA | 39.20 | NA | 5.67 | 2.37 | NA | 1.17 | 0.82 | NA | 1.17 |
| Int. F.: 1650 Hz | 0.04 | 0.12 | 0.03 | 0.01 | 0.01 | NA | 0.04 | 0.05 | NA | 0.03 | 0.10 | NA |
| Int. F.: 2610 | 0.00 | 0.02 | 0.03 | 0.02 | 0.01 | 0.00 | 0.45 | 0.17 | 0.06 | 0.05 | 0.08 | 0.02 |
| Har. F.: 3000 | 0.10 | NA | NA | 0.02 | NA | 0.01 | 0.23 | NA | 0.04 | 0.06 | 1.50 | 0.11 |
| Har. F.: 3780 | 0.02 | NA | NA | 0.00 | NA | NA | 0.02 | NA | 0.17 | 1.11 | 0.04 | NA |
| Fund. 60 A.: 1 | 1.70 | 41.22 | 73.56 | 2.04 | 87.68 | 0.06 | 2.65 | 2.28 | 0.00 | 1.21 | 5.17 | 0.23 |
| Int. 45 A.: 0.2 | 11.50 | NA | NA | 11.20 | NA | 141.30 | 15.75 | NA | 7.31 | 2.65 | NA | 6.75 |
| Int. 1650 A.: 0.3 | 7.47 | 15.70 | 32.90 | 0.53 | 4.27 | NA | 4.60 | 13.80 | NA | 4.00 | 7.27 | NA |
| Int. 2610 A.: 0.2 | 7.35 | 51.70 | 78.20 | 2.55 | 0.55 | 6.50 | 11.80 | 3.00 | 88.17 | 1.20 | 32.85 | 39.67 |
| Har. 3000 A.: 0.1 | 10.20 | NA | NA | 2.40 | NA | 20.20 | 7.00 | NA | 76.85 | 1.10 | 15.70 | 74.90 |
| Har. 3780 A.: 0.1 | 7.00 | NA | NA | 3.10 | NA | NA | 6.80 | NA | 184.90 | 8.00 | 7.00 | NA |
| Fund. 60 $\phi$: 25° | 7.21 | 54.74 | 27.91 | 9.50 | 35.11 | 88.80 | 2.60 | 33.36 | 131.10 | 0.27 | 20.53 | 154.10 |
| Int. 45 $\phi$: 30° | 6.99 | NA | NA | 11.55 | NA | 5.15 | 4.70 | NA | 38.11 | 4.22 | NA | 8.79 |
| Int. 1650 $\phi$: 45° | 4.60 | 4.84 | 17.27 | 0.90 | 9.89 | NA | 1.21 | 24.19 | NA | 9.48 | 28.29 | NA |
| Int. 2610 $\phi$: 60° | 7.85 | 24.63 | 9.34 | 11.73 | 12.31 | 3.50 | 7.12 | 38.83 | 72.12 | 4.49 | 4.32 | 23.56 |
| Har. 3000 $\phi$: 68° | 3.70 | NA | NA | 1.09 | NA | 7.65 | 0.04 | NA | 41.62 | 5.97 | 52.29 | 33.40 |
| Har. 3780 $\phi$: 50° | 4.70 | NA | NA | 0.59 | NA | NA | 7.84 | NA | 19.58 | 2.44 | 1.54 | NA |

The analysis of the simulation results in Tables 7–9 is presented in the following subsections.

### 4.2.1. 45 Hz Sub-Harmonic Parameters' Estimation

For the $s_2(t)$ signal contaminated with pink noise, while FB-ESPRIT detected the 45 Hz sub-harmonic component at 5 dB SNR, the conventional ESPRIT and MUSIC methods were not able to detect such a component, and even the MUSIC technique was not able to detect the component also for an $SNR = 20$ dB. Table 8 shows a clear improvement in the estimates of the frequency, amplitude, and phase parameters for the proposed method compared to the methods under analysis. According to the power spectral density curve (Figure 5) of the pink noise, it was expected that there would be a larger disturbance for lower frequencies, such as sub-harmonics. Thus, the proposed method suffered a greater variation in the estimation of its parameters at $SNR = 5$ dB, but when compared with the ESPRIT technique and MUSIC, this variation was much smaller, while FB-ESPRIT outperformed. For example: in ESPRIT, the mean of the measured errors at 20 dB and 40 dB for the frequency, amplitude, and phase parameters was approximately 6.15%, 12.31%, and 96.47%, respectively.

Table 9 indicates the results on the signal $s_2(t)$ contaminated with 5 dB SNR colored noises of pink, red, blue, and violet. It is observable that just FB-ESPRIT was able to detect the 45 Hz sub-harmonic components, but not ESPRIT and MUSIC. Even in the cases that the signal was contaminated by pink and red noises, as these noises were the most conflicting noises due to their power spectral density curves, FB-ESPRIT detected 45 Hz (Figure 5).

### 4.2.2. 60 Hz Fundamental Parameters Estimation

In evaluating the 60 Hz fundamental component parameter estimation over signal $s_2(t)$ contaminated by pink noise, FB-ESPRIT obtained good estimates for frequencies, amplitudes, and phases with a maximum error in the phase estimation of approximately 7.21%. In contrast, the conventional ESPRIT and MUSIC methods had high error estimates, mainly in the amplitudes and phases. In the case of the ESPRIT technique, the mean of the measured errors of frequency, amplitude, and phase parameters was approximately 0.63%, 12.85%, and 28.09%, respectively, for the noise range of 5 dB to 40 dB; in MUSIC, it was approximately 0.54%, 25.75%, and 19.20%, respectively, for frequency, amplitude, and phase estimations; whereas for FB-ESPRIT, it was approximately 0.31%, 1.56%, and 2.58%, respectively. Observe Figure 13, which shows the percent error of frequency, amplitude, and phase of this component indicating how the proposed method excelled in comparison with the other methods in estimating this component.

In addition to the pink noise, considering the $s_2(t)$ signal contaminated by the red, blue, and violet noises of an SNR of 5 dB, it is observed that FB-ESPRIT obtained relevant estimates with a maximum error in phase estimation for the signal contaminated with red noise of about 9.50%. Table 9 shows that the results of the other methods under analysis were poor even with an error above 100% for amplitude estimation when the signal was contaminated with red noise for the MUSIC technique. The ESPRIT technique had a high estimate error of 87.68% under pink noise.

### 4.2.3. Harmonics 3000 Hz and 3780 Hz and Inter-Harmonics 1650 Hz and 2610 Hz Parameters' Estimation

For the $s_2(t)$ signal contaminated with pink noise considering the frequency estimates for the 1650 Hz and 2610 Hz interharmonic components and the 3000 Hz and 3780 Hz harmonic components, the three methods obtained reasonable estimates, but for the amplitude and phase estimates, conventional ESPRIT and MUSIC failed and obtained high errors. Moreover, the MUSIC method was not able to detect the 3000 Hz and 3780 Hz harmonic components and the ESPRIT method the 3780 Hz harmonic component for an SNR of 5 dB.

Considering the 1650 Hz interharmonic, it is observed that the proposed method obtained not only good estimates for frequencies, but also for amplitudes and phases, obtaining a maximum error of approximately 7.47% in the amplitude estimation with $SNR = 5$ dB. In contrast, the amplitude and

phase estimates' errors of the confronted methods were higher, and this result can be seen in Figure 14, which shows the percentage error of the frequency, amplitude, and phase of this component, indicating a better calculation of the proposed method for the estimated parameters of this component. In the case of signal $s_2(t)$ contaminated with noise other than pink noise at an SNR of 5 dB, the proposed method also outperformed the two other methods, and the worst result came from the MUSIC technique, which presented high error estimates for amplitudes and phases.

For the 2610 Hz interharmonic, the proposed method maintained good estimates compared to the others. Figure 15 shows the percentage error of the frequency, amplitude, and phase of this component and indicates how the proposed method was better than the other techniques in the estimation of this component. In the case of signal $s_2(t)$ contaminated with colored noise other than pink at an SNR of 5 dB, the proposed method indicated a variation in frequency estimation when there was contamination with blue noise, which may be related to the filter bank configuration. Remember that the filter bank was not ideal, but a uniform equally spaced one. However, it is worth mentioning that in general, for the estimation of this component, FB-ESPRIT stood out again as the best method for the estimation under colored noise.

For the 3000 Hz harmonic, the ESPRIT and MUSIC techniques had more significant errors than FB-ESPRIT with greater emphasis on the MUSIC technique, which obtained error above 100% in the estimation of the component amplitude. Moreover, neither technique was not able to estimate the components at an SNR of 5 dB. For the $s_2(t)$ signal contaminated with noise other than pink at an SNR of 5 dB, the other techniques again failed to estimate this component, and the highlight for a good estimate went to FB-ESPRIT, which was able to detect the component at all levels of SNRs, while the maximum estimation error was in the amplitude parameter 10.20% with the signal contaminated by pink noise.

For the 3780 Hz harmonic, the proposed method had good estimation results. Note that this harmonic was much closer to the sampling range limitation, and yet, FB-ESPRIT was able to detect the components and estimate the parameters with reasonable errors even for an SNR of 5 dB. In contrast, for the same $SNR$, neither ESPRIT nor MUSIC were able to detect this component. In the case of the $s_2(t)$ signal contaminated with noise other than pink at an SNR of 5 dB, the ESPRIT technique only detected the component when the signal was contaminated with violet noise, and MUSIC suffered from high errors above 100%. The proposed method detected the component with reasonable error estimates (see Table 9).

### 4.3. Evaluation of a Photovoltaic Power Plant Signal

A practical application with colored noise is in a photovoltaic (PV) power plant. The colored noise in the PV power signal is indeed the low frequency interferences originating from the oscillations in sunlight emission to the solar panels mainly due to windy or cloudy weather. We compared FB-ESPRIT's efficiency with ESPRIT on a PV signal measured from the PV power plant of the solar laboratory at the Federal University of Juiz de Fora (UFJF), Brazil, as shown in Figure 16. The low frequency interference in the PV power signal was due to the partly cloudy and windy weather at the time of measuring the power signal. Figure 17 shows 200 ms of the PV signal and its FFT spectrum, wherein the low frequency oscillation besides the fundamental of 60 Hz is clearly observable. In addition, from the FFT spectrum, we can observe a kind of sparse interference in the low frequency range of [0–20] Hz. This interference resembles a colored noise and can affect the performance of a parametric spectral estimation such as ESPRIT.

To have a comparative evaluation of this real case, we applied both ESPRIT and FB-ESPRIT to estimate the PV power signal components' parameters. Table 10 shows the estimated parameters for the components of the PV current signal by ESPRIT and FB-ESPRIT. The first interesting observation is that while FB-ESPRIT detected five components and estimated their parameters, ESPRIT detected just two components. To have a numerical evaluation of the estimated parameters, since we did not have reference values for the estimated parameters, we reconstructed the power signal by the estimated

parameters and numerically compared the reconstructed power signals to the real one. Figure 18 shows the reconstructed power signals using the estimated parameters and visualizing the extent that each follows the original real power signal. As is observable from Figure 18a, the FB-ESPRIT estimated and reconstructed signal (solid-blue) followed the original signal closely (dashed-black), while in the case of ESPRIT, it can be observed in Figure 18b that the reconstructed signal was away from the real one, especially in the part highlighted by the ellipse. Furthermore, the mean absolute error of the reconstructed signals from the original one indicated higher accuracy of the estimated parameters by FB-ESPRIT as it was 8.3844 and 13.6845 for FB-ESPRIT and ESPRIT, respectively.

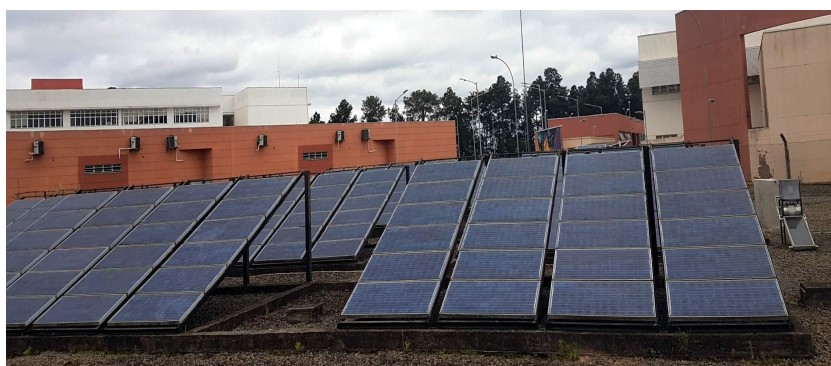

**Figure 16.** Photo-voltaic power plant of the solar laboratory at Federal University of Juiz de Fora (UFJF), Brazil.

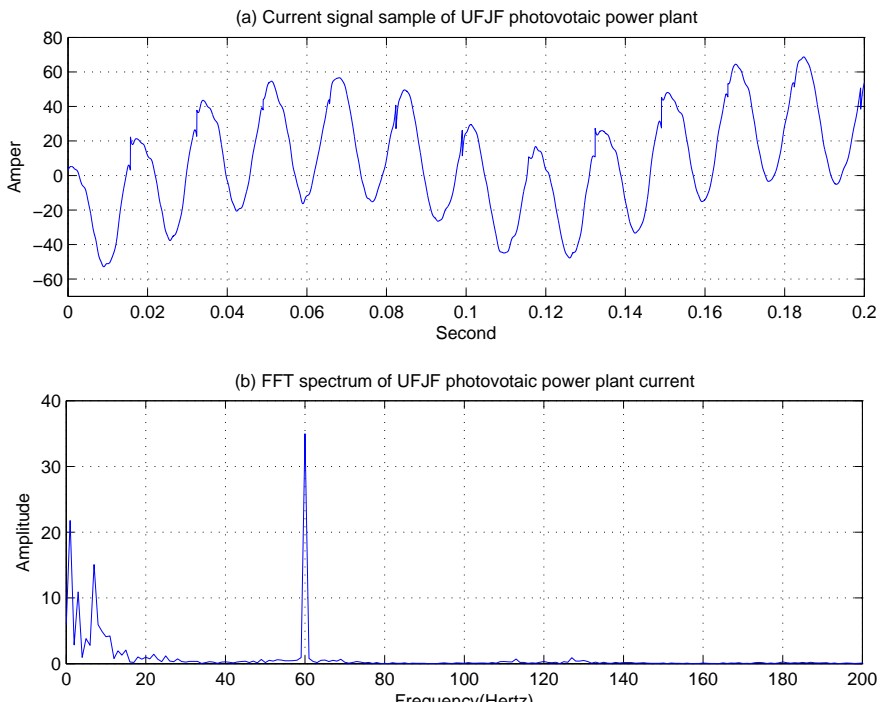

**Figure 17.** (**a**) Two hundred milliseconds of the PV signal measured from the UFJF PV power plant and its (**b**) FFT spectrum with colored interference in the frequency range of [0–20] Hz.

**Table 10.** Estimated parameters of the UFJF PV power plant current signal components by (a) ESPRIT and (b) FB-ESPRIT.

| (a) By ESPRIT | | | |
|---|---|---|---|
| **Components** | **Frequency** | **Amplitude** | **Phase** |
| #1 | 7.3850 Hz | 28.7338 | $-159.2012°$ |
| #2 | 60.0511 Hz | 33.6139 | $25.9615°$ |
| (b) By FB-ESPRIT | | | |
| **Components** | **Frequency** | **Amplitude** | **Phase** |
| #1 | 1.0317 Hz | 20.1455 | $-26.5236°$ |
| #2 | 3.8457 Hz | 2.0252 | $97.0703°$ |
| #3 | 7.0823 Hz | 18.3540 | $-129.7700°$ |
| #4 | 10.1324 Hz | 5.1732 | $117.0557°$ |
| #5 | 59.9769 Hz | 33.7309 | $34.0749°$ |

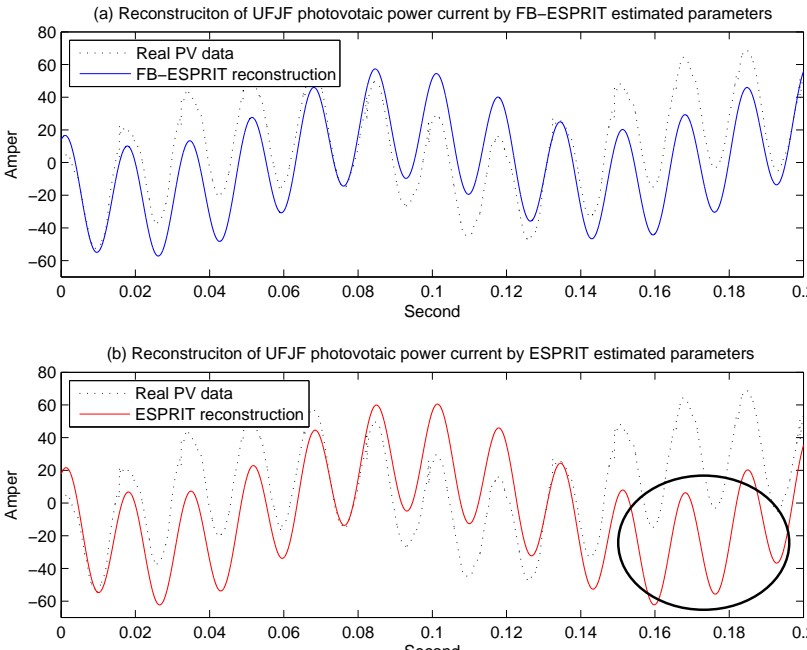

**Figure 18.** Original power signal (dashed-black) and the reconstructed signals using the parameters estimated by (**a**) FB-ESPRIT (solid-blue) and (**b**) ESPRIT (solid-red).

## 5. Conclusions

The filter bank associated ESPRIT improved the estimation efficiency compared to the ESPRIT technique, demonstrating higher accuracy and speed in the estimation. In addition to the above mentioned advantages in the literature, this manuscript studied and analyzed the robustness of FB-ESPRIT to colored noise of different types. While parametric sub-space methods assume whiteness for the contaminating noise, FB-ESPRIT, as expected and evaluated, was robust to colored noise. This is because of it being inherent to the filter bank and spread spectrum process, which the first one dividing the spectrum into sub-bands and the latter one extended each sub-band spectrum to the full width of the signal bandwidth. Therefore, (i) the spectrum of the noise at each sub-band resembled a more monotonic spectrum and its whiteness increased, and (ii) the energy of the noise was divided into the number of sub-bands. The simulation results under four colors of noise, pink, red, blue, and violet, demonstrated the considerable superiority of FB-ESPRIT to ESPRIT and MUSIC. Furthermore,

a practical application of FB-ESPRIT with a comparison to ESPRIT in the spectral estimation of the photo-voltaic power signal under the colored low frequency interferences was presented.

**Author Contributions:** Conceptualization, M.K., C.A.D., A.S.C. and M.A.A.L.; methodology, E.S., M.K., A.S.C., C.A.D. and M.A.A.L.; software, E.S., M.K. and M.A.A.L.; validation, C.A.D., A.S.C. and A.Y.; formal analysis, C.A.D., A.S.C. and A.Y.; investigation, E.S. M.K. and M.A.A.L.; resources, C.A.D. and A.S.C.; data curation, E.S.; writing–original draft preparation, C.A.D. and A.S.C.; writing–review and editing, E.S. and M.K.; supervision, C.A.D. and A.S.C.; project administration, C.A.D. and A.S.C.; funding acquisition, E.S., M.K., A.S.C., C.A.D., M.A.A.L., and A.Y.

**Funding:** Brazillian funding agencies CAPES, CNPq, FAPEMIG and INERGE.

**Acknowledgments:** The authors are grateful to the anonymous referees and thank them for their helpful and constructive comments. They would equally like to extend thanks to the Ministries of Education and Science of the Brazil and Japan for providing stimulating environments towards the realization of this research. Also, special thanks go to Brazilian funding agencies CAPES, CNPq, FAPEMIG and INERGE for funding this research.

**Conflicts of Interest:** The authors declare no conflict of interest.

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
