# Peer review of "High Accuracy Power Quality Evaluation under a Colored Noisy Condition by Filter Bank ESPRIT"

_electronics, doi:10.3390/electronics8111259_

Round 1
Reviewer 1 Report
This paper presents an estimation of signal parameters via rotational invariance technique (ESPRIT) for measuring the harmonics and inters harmonics in power systems. To show the superiority of the proposed scheme the authors also compared the performance with existing schemes. By the way, the reviewer could not find any technical contribution to this manuscript. However, the following remarks should be addressed in order to improve the present manuscript.
First of all, the presentation of this paper is very poor. The contribution of this paper is not clear i.e., what is the main contribution of the paper compared to the existing literature. The abstract of this paper should be rewritten because it is not clear to the reviewer at all. The reviewer could not find any significance of Fig. 1. The authors highlighted the applications of the proposed scheme on power systems. However, this is not reflected in the simulation section. Finally, there are lots of grammatical errors and typos. Please thoroughly check the whole paper to correct all of these errors.Author Response
Before everything thing, we thank a lot the reviewer for the detailed comments.
The responses to the reviewer’s general and major points have been provided after each point.
General comments:
Suggestions for Authors
This paper presents an estimation of signal parameters via rotational invariance technique (ESPRIT) for measuring the harmonics and inters harmonics in power systems. To show the superiority of the proposed scheme the authors also compared the performance with existing schemes. By the way, the reviewer could not find any technical contribution to this manuscript. However, the following remarks should be addressed in order to improve the present manuscript.
Authors response:
Thanks a lot for the detailed view and your concern on the manuscript. The major points of your comments are listed below and responded.
Point 1: First of all, the presentation of this paper is very poor.
Authors Response:
Following the worthwhile comments of the Reviewer, we have improved the presentation of the paper as we tried to clear the contribution of the paper.
It has been explained in detail the tasks for improvement of the presentation and clarification of the contribution in the response to Point 2 (below).
Point 2: The contribution of this paper is not clear i.e., what is the main contribution of the paper compared to the existing literature.
Authors Response:
The main contribution of the manuscript is first pointing to a technical problem due to the colored noise in power systems especially in the case of renewable sources like wind turbines. The parametric spectrum estimation techniques used in literature for power quality analysis are all by the assumption of the Gaussianity of the noise, thus their efficiency is degraded as the noise is colored. Second, the manuscript suggests FB-ESPRIT as an efficient high accuracy technique for power quality analysis at the presence of colored noise since it overcomes the colored noises of different types. The manuscript illustrates how FB-ESPRIT mitigates the colored noise, then it approves it via the simulation results on synthetic power line signals with a complicated spectrum containing a fundamental 60Hz, subharmonic of 30Hz, and several harmonics and interharmonics under contamination of different SNR levels of additive colored noise of different types.
The contribution of the paper has been discussed well in the revised paper and has been addressed during the paper with more clear explanations as below. It is highlighted in blue inside the revised paper for your observation.
All the abstract has been rewritten. On page 1, Introduction, lines 26-31, the colored noise as a practical issue in the case of Renewables and the result of the proliferation of new power electronics. In the Introduction Section, On page 2, lines 56-58, the susceptibility of ESPRIT to colored noise has been mentioned. While ESPRIT is a well-known high-resolution parametric technique for power quality analysis by spectral components parameters estimation, it is the same as all other parametric techniques based on assumption white Gaussian additive noise, and its efficiency is degraded at the presence of colored noise. In the Section `2 Colored Noise and Parametric Estimation’, On Page 3, lines 91-99, the reason of dependency of the parametric subspace division techniques for power spectrum analysis to the Gaussianity of the noise has been explained, and also it has been clarified why colored noise makes problem to these techniques such as ESPRIT. The Section ‘FB-ESPRIT versus colored noise’ and Figure 4 discusses why FB-ESPRIT should be less affected by the colored noise. It illustrates how the effect of filter bank followed by the down-sampling increases the monotony of each sub-band and helps the individual ESPRIT applied to each sub-bands performs with higher efficiency. (Page7, lines 142-158)
Point 3: The abstract of this paper should be rewritten because it is not clear to the reviewer at all.
Authors Response:
Thanks a lot for the worthwhile comment.
Following your comments for more clarity on the article contributions, we have rewritten the abstract of the manuscript. The revised abstract of the manuscript is as follows:
“Due to the highly increasing integration of renewable energy sources to the power grid and their fluctuations besides the recent growth of new power electronics equipment, the noise in power systems has become colored. The colored noise affects the methodologies for power quality parameters estimation, such as harmonics and interharmonics components. Estimation of signal parameters via rotational invariance techniques (ESPRIT) as a parametric technique with high resolution has proved its efficiency in the estimation of power signal components frequencies, amplitudes, and phases for quality analysis, under the assumption of white Gaussian noise. Since ESPRIT suffers from the high computational effort, the filter banked ESPRIT (FB-ESPRIT) was suggested for mitigation of the complexity. This manuscript suggests the FB-ESPRIT as well for accurate and robust estimation of power signal components parameters at the presence of the colored noise. Even though the parametric techniques depend on the Gaussianity of contaminating noise to perform properly, FB-ESPRIT performs well in colored noise. The FB-ESPRIT superiority compared with conventional ESPRIT and MUSIC techniques has been demonstrated through many simulations runs on synthetic power signals with multiple harmonics, inter-harmonics, and subharmonic components at the presence of noises of different colors and different SNR levels. FB-ESPRIT has a significant efficiency superiority in power quality analysis with a wide gap distance from the other estimators, especially under the high level of colored noise.”
Point 4: The reviewer could not find any significance of Fig. 1.
Authors Response:
Figure 1 was just illustrating the spectrum of white noise and it was given besides the spectral density of colored noises to give a comparative view to the reader. However, the monotony of the white noise spectrum is a well-known issue, and following your comment, we removed the Figure 1 in the revised version of the manuscript.
Point 5: The authors highlighted the applications of the proposed scheme on power systems.
However, this is not reflected in the simulation section.
Authors Response:
Normally a power line signal contains a fundamental component of 60Hz, and some harmonics. To have a comprehensive evaluation of colored noise effect on the estimation of the signal parameters of frequency, amplitude, and phase for all the signal components, we performed the evaluative simulations over two synthesized power signals containing a sub-harmonic, multiple harmonics and interharmonics in addition to the fundamental component. The synthetic power signals represent a complex set of parameters where each is differently affected by colored noise.
The analytical comparison is between the performance of FB-ESPRIT and the conventional ESPRIT in the estimation of the parameters of harmonics, inter-harmonics, and subharmonics of the synthesized power signals under different colored noise conditions.
Following your comment for more clarity about this issue, we have added the above explanation at the opening of the results and discussions in the revised manuscript as it is observable in lines 160-167.
Also, the following explanation has been added before the formulation of first case of study, as it is observable in the revised manuscript in lines 176-177 highlighted in blue.
``As a power signal with a complex content of components, a synthesized sinusoidal signal with multiple harmonics, inter-harmonics, and sub-harmonic components is acquired as the case study of parameter estimation for evaluation and comparison of FB-ESPRIT with conventional ESPRIT at the presence of colored noise.’’
As well, in lines 294-295, the following explanation has been added as it is highlighted in blue in the revised manuscript:
``The synthesized power signal s_2(t) in Equ. (22) contains a 60Hz fundamental, a sub-harmonic , two inter-harmonics, and two harmonic components.’’
Point 6: Finally, there are lots of grammatical errors and typos. Please thoroughly check the whole paper to correct all of these errors.
Authors response:
Thanks a lot for your notice and indication.
To avoid grammatical errors and typos, we have overchecked the manuscript after revision having also the technical view of our English native colleagues.
We believe that the current revised version of the manuscript has no English error.

Reviewer 2 Report
What is 2,7 in galactic noise? Explain cte on line 76 The paper started off with power quality and after doing signal processing the topic power quality is forgotten. How is the whole body of research applied to power quality?Author Response
Dear Reviewer 2
Before everything thing, we thank you a lot for the worthwhile comments.
The response to the general and major points has been provided in detail in the attached PDF.
Please observe it.
Thanks and Regards

Round 2
Reviewer 1 Report
If the authors can show any results of the proposed scheme in applications of power systems then it should be OK to publish this manuscript in this paper.
Author Response
Before everything thing, we thank you again for your constructive comments.
It helped us a lot to enrich the paper.
Comments:
If the authors can show any results of the proposed scheme in applications of power systems then it should be OK to publish this manuscript in this paper.
Authors response:
Thanks a lot for your worthwhile suggestion. Following your comment, a practical application with colored noise for photovoltaic (PV) power plant has been added to the paper.
The colored noise in the PV power signal is indeed the low-frequency interferences originated from the oscillations in sunlight emission to the solar panels mainly due to windy cloudy weather. We have compared the FB-ESPRIT efficiency with ESPRIT on PV signal measured from the PV power plant of the solar laboratory in the Federal University of Juiz de Fora (UFJF), Brazil. The low-frequency interference in the PV power signal is due to the partly cloudy windy weather at the time of measuring the power signal.
To have a comparative evaluation on this real case, we have applied both ESPRIT and FB-ESPRIT to estimate the PV power signal components parameters. Table 10 added which shows the estimated parameters for the components of the PV current signal by ESPRIT and FB-ESPRIT. The first interesting observation is that while the FB-ESPRIT detects five components and estimates their parameters, ESPRIT detects just two components.
To have a numerical evaluation of the estimated parameters, since we do not have reference values for the estimated parameters, we reconstruct the power signal by the estimated parameters, and numerically compare the reconstructed power signals by the real one. Fig. 18 has been added which shows the reconstructed power signals using the estimated parameters and visualized the extent that each follows the original real power signal. As it is observable from Figure the FB-ESPRIT estimated and reconstructed signal follows closely the original signal, while in the case of ESPRIT, the reconstructed signal takes distance from the real one especially in the part highlighted by the ellipse. Also, the mean absolute error of the reconstructed signals from the original one indicates higher accuracy of the estimated parameters by FB-ESPRIT as it is 8.3844, and 13.6845 for FB-ESPRIT and ESPRIT respectively.
The corresponding revision can be found in the revised manuscript on Pages 19 -20, lines 376-399, Figures 16, 17, and 18, and Table 10. In conclusion, lines 410-412.
